# Deep phenotyping of post-infectious myalgic encephalomyelitis/chronic fatigue syndrome

Post-infectious myalgic encephalomyelitis/chronic fatigue syndrome (PI-ME/CFS) is a disabling disorder, yet the clinical phenotype is poorly defined, the pathophysiology is unknown, and no disease-modifying treatments are available. We used rigorous criteria to recruit PI-ME/CFS participants with matched controls to conduct deep phenotyping. Among the many physical and cognitive complaints, one defining feature of PI-ME/CFS was an alteration of effort preference, rather than physical or central fatigue, due to dysfunction of integrative brain regions potentially associated with central catechol pathway dysregulation, with consequences on autonomic functioning and physical conditioning. Immune profiling suggested chronic antigenic stimulation with increase in naïve and decrease in switched memory B-cells. Alterations in gene expression profiles of peripheral blood mononuclear cells and metabolic pathways were consistent with cellular phenotypic studies and demonstrated differences according to sex. Together these clinical abnormalities and biomarker differences provide unique insight into the underlying pathophysiology of PI-ME/CFS, which may guide future intervention.

Myalgic encephalomyelitis/chronic fatigue syndrome (ME/CFS) is the commonly used term to describe a disorder of persistent and disabling fatigue, exercise intolerance, malaise, cognitive complaints, and physical symptoms with significant socioeconomic consequences. The physiological mechanism responsible for the persistence of fatigue and related symptoms has yet to be determined. Immunologic[1], bioenergetic[2–5], and physiologic[6] alterations have been reported. However, many of these findings have been inconsistent and both their clinical relevance and relation to each other are unclear. Hence, there are currently no effective disease modifying treatments for ME/CFS, and even developing and testing of potential new treatments is hampered by difficulty in defining cases or tracking response through symptoms or biomarkers.

A major obstacle to rigorous ME/CFS research is case assessment due to the absence of a diagnostic biomarker. Over 20 different diagnostic criteria underscore the difficulty in defining the clinical symptom content of ME/CFS[7]. The usual symptoms are non-specific and overlap with other diseases, hence misattribution is frequent and ME/CFS is typically a diagnosis of exclusion[8]. ME/CFS often occurs following an acute infection, designated post-infectious-ME/CFS

(PI-ME/CFS), with an estimated incidence of 10–12% after certain infections[9,10]. The most well-known association of PI-ME/CFS has been with the Epstein-Barr Virus[11] but as the full extent of the sequelae of the COVID-19 pandemic are better understood, SARS-CoV-2 may become an even stronger correlate[12].

In 2016, the National Institutes of Health (NIH) launched an initiative to study ME/CFS. The NIH Division of Intramural Research developed an exploratory clinical research program to perform deep phenotyping on a cohort of PI-ME/CFS participants and healthy volunteers (HV) as controls. Prior to the SARS-CoV-2 pandemic, this study recruited a cohort of well-characterized PI-ME/CFS patients and applied modern broad and deep scientific measures to describe their biophenotype compared to HVs. The aim was to identify relevant group differences that could generate new hypotheses about the pathogenesis of PI-ME/CFS and provide direction for future research. Over 75 scientists and clinicians across 15 of the 27 institutes that comprise the NIH contributed to this multi-disciplinary work. Importantly, we developed rigorous inclusion criteria which comprised detailed medical and psychological evaluations to minimize diagnostic misattribution. A relatively homogenous population was recruited in

✉ e-mail: Avindra.nath@nih.gov

whom symptoms were initiated after infection. This study aimed to investigate the underlying pathophysiological mechanisms. The participants underwent a multi-dimensional evaluation that included a wide range of physiological measures, physical and cognitive performance testing, and biochemical, microbiological, and immunological assays of blood, cerebrospinal fluid, muscle, and stool. Measurement techniques were developed to query issues such as physical capacity, effort preference, and deconditioning that may confound the results. Multi-omic measurements of gene expression, proteins, metabolites, and lipids were performed in parallel on collected samples. This report summarizes and contextualizes the main findings arising from this work.

## Results

Data are reported in the following formats: (HV mean ± standard deviation versus PI-ME/CFS mean ± standard deviation, $p$-value). Odds ratio (OR) and relative odds ratio (ROR) are reported as: HV:PI-ME/CFS ratio [95% confidence interval]. Additional analysis and a glossary of abbreviations are provided in the Supplementary Information.

### Case ascertainment

Study recruitment occurred between December 2016 and February 2020 (Fig. 1a). Of 484 ME/CFS inquiries, 217 individuals underwent detailed case reviews entailing telephone interviews and medical record review (Supplementary Data S2). Of these, 27 underwent in-person research evaluation and 17 were determined to have PI-ME/CFS by a panel of clinical experts with unanimous consensus. Twenty-one comparator healthy volunteers were recruited separately. Additional recruitment was terminated due to the COVID-19 pandemic.

### Cohort characteristics

Characteristics of the participants are detailed in Fig. 1b, c and Supplementary Data S5; there were no group differences in age, sex, or BMI. There were no group differences in performance validity testing (Supplementary Data S6); a method for determining whether neuropsychological test performances are overly impacted by non-cognitive factors[13]. Patient reported outcome measurement information system (PROMIS) and other questionnaires for fatigue, emotional distress, sleep disturbances, anxiety and other symptoms were greater ($p < 0.05$) in PI-ME/CFS than HVs (Fig. 1c, Supplementary Data S7). There were no clinically relevant findings on physical examination, psychological evaluation, or laboratory testing in either group (Supplementary Data S8, 9). Measurements of small nerve fiber density and neuronal injury markers in blood and cerebrospinal fluid were not different between groups (Supplementary Fig. S1). Polysomnography did not reveal clinically relevant findings (Supplementary Information, Sleep dysfunction). The groups did not differ in composition by whole-body dual energy X-ray absorptiometry or in slow-to-fast muscle fiber atrophy[14] as measured by Type 2: Type 1 muscle fiber median Feret diameter ratio (Type2:1 mFd) using ATPase pH 9.4 stain of the vastus lateralis (Supplementary Figs. S2 and S3D).

### Autonomic dysfunction as measured by multiple parameters in PI-ME/CFS

Head-up tilt table testing for up to 40 min showed no group differences in frequency of orthostatic hypotension, excessive orthostatic tachycardia, or tilt-related symptoms requiring test cessation. Twenty-four hour ambulatory ECG showed that PI-ME/CFS participants had diminished heart rate variability (HRV) in SDNNi, rMSSD, and pNN50 time domain indices (Fig. 2a–d). PI-ME/CFS participants also had altered frequency domain differences, with decreased high frequency (HF) and low frequency (LF) power (Fig. 2e, f). Non-linear analyses adjusted by hour of the day further demonstrated group differences in ($p = 1.1E-12$), SD2 ($p = 2.1E-05$), and SD1/SD2 ($p = 3.9E-07$). Similarly, group heart rates displayed two notable trends (Fig. 2g). Increased

heart rate in PI-ME/CFS participants throughout the course of a day suggests comparatively increased sympathetic activity. PI-ME/CFS participants also had a diminished drop in nighttime heart rate suggesting diminished parasympathetic activity, consistent with observed differences in rMSSD, HF power, pNN50, and SD1. A decrease in baroslope (Fig. 2h) and longer blood pressure recovery times after the Valsalva maneuver ($3.0 \pm 0.2$ versus $4.1 \pm 0.4$ sec, $p = 0.014$) in PI-ME/CFS reflect decreased baroreflex-cardiovagal function. Considered together, these data suggest that there is an alteration in autonomic tone, implying central nervous system regulatory change.

### Altered effort preference in PI-ME/CFS

The Effort-Expenditure for Rewards Task (EEfRT)[15] was used to assess effort, task-related fatigue, and reward sensitivity (Supplementary Fig. S5A–D). Given equal levels and probabilities of reward, HVs chose more hard tasks than PI-ME/CFS participants (Odds Ratio (OR) = 1.65 [1.03, 2.65], $p = 0.04$; Fig. 3a). Two-way interactions showed no group differences in responses to task-related fatigue (Relative Odds Ratio (ROR) = 1.01 [0.99, 1.03], $p = 0.53$; Fig. 3a), reward value (ROR = 1.57 [0.97, 2.53], $p = 0.07$), reward probability (ROR = 0.50 [0.09, 2.77], $p = 0.43$), and expected value (ROR = 0.98 [0.54, 1.79], $p = 0.95$). Effort preference, the decision to avoid the harder task when decision-making is unsupervised and reward values and probabilities of receiving a reward are standardized, was estimated using the Proportion of Hard-Task Choices (PHTC) metric. This metric summarizes performance across the entire task and reflects the significantly lower rate of hard-trial selection in PI-ME/CFS participants (Fig. 3a). There was no group difference in the probability of completing easy tasks but there was a decline in button-pressing speed over time noted for the PI-ME/CFS participants (Slope = −0.008, SE = 0.002, $p = 0.003$; Fig. 3b). This pattern suggests the PI-ME/CFS participants were pacing to limit exertion and associated feelings of discomfort[16]. HVs were more likely to complete hard tasks (OR = 27.23 [6.33, 117.14], $p < 0.0001$) but there was no difference in the decline in button-press rate over time for either group for hard tasks (Fig. 3b).

### Equivalent motor performance in PI-ME/CFS and healthy volunteers

On single grip task, maximal voluntary contraction (MVC) was not different between groups (Fig. 3c) and correlated with lean arm mass but not effort preference or Type2:1 mFd (Supplementary Fig. S6A–C). Time to failure, the inability to maintain grip force at 50% of maximum contraction, was significantly shorter ($p = 0.0002$) in the PI-ME/CFS participants (Fig. 3d) and correlated with effort preference (Fig. 3e) in PI-ME/CFS but not in HVs. Time to failure did not correlate with lean arm mass or Type2:1 mFd in either group.

Repetitive grip testing was performed on a subgroup of participants. MVC did not differ between groups. A rapid decline in force along with a significantly lower number of non-fatigued blocks (Fig. 4a) and a relative decrease in the slope of the Dimitrov index[17,18] (Fig. 4b) occurred in PI-ME/CFS participants but both remained constant in HVs, suggesting that the decline of force was not due to peripheral fatigue or a neuromuscular disorder. Motor Evoked Potential amplitudes using transcranial magnetic stimulation of HVs decreased over the course of the task, consistent with post-exercise depression as seen in healthy and depressed volunteers[19], while they increased in PI-ME/CFS participants (Fig. 4c). This indicates that the primary motor cortex remained excitable for PI-ME/CFS, suggesting reduced motor engagement from this group[20].

### Decreased activation of integrative brain regions in PI/ME-CFS

Repetitive grip testing by functional brain imaging was performed on a subgroup of participants and also showed a rapid decline in force (Supplementary Fig. S7E) and a significantly lower number of non-fatigued blocks ($p = 0.004$) in PI-ME/CFS. First, we assessed commonly

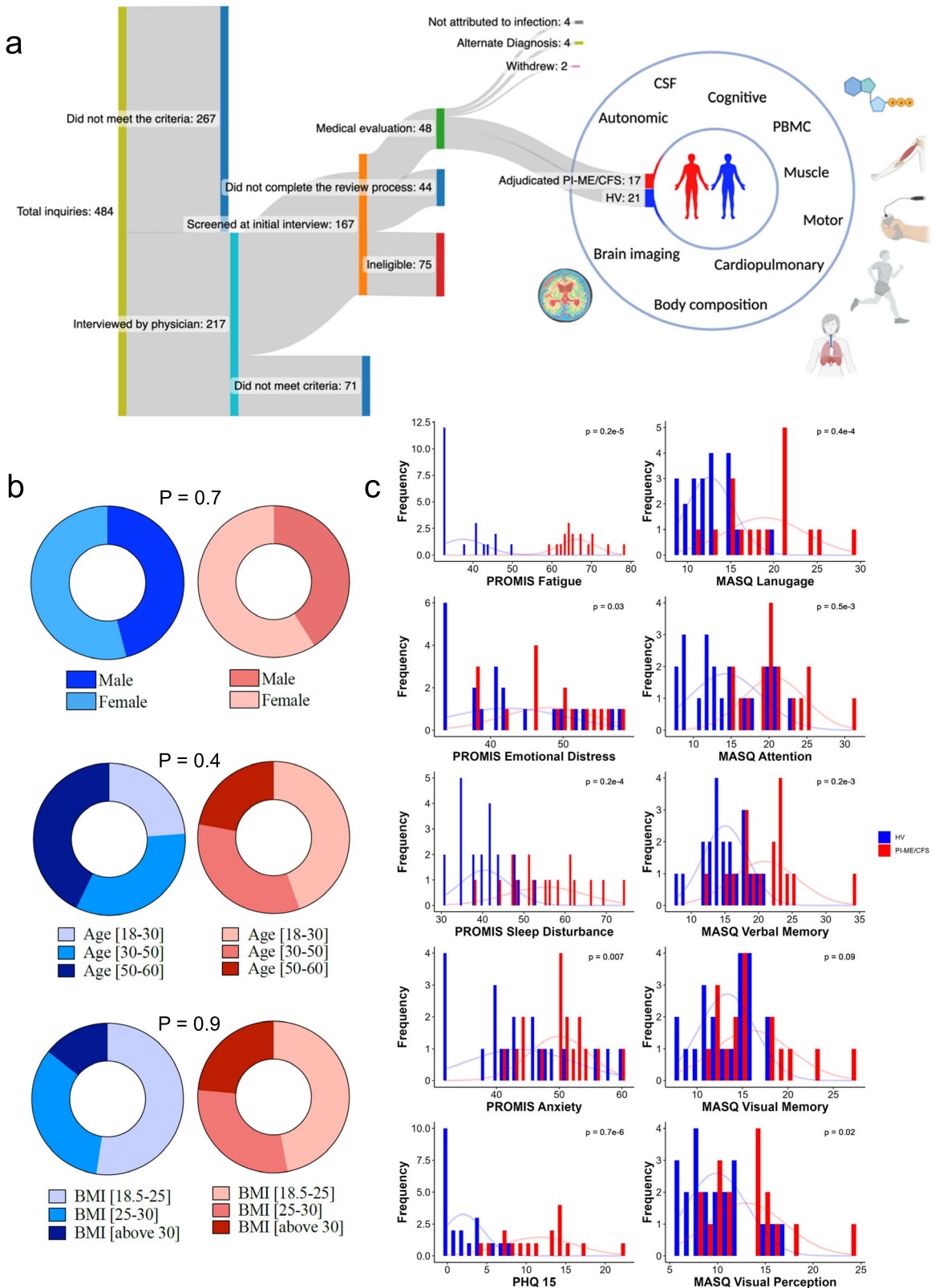

activated brain areas by implementing a conjunction analysis in which we took each group t-test across all blocks, thresholded voxels at $p \leq 0.01$ with a multiple comparison correction at p = 0.05, $k > 65$ and kept voxels that were commonly activated in each group. HV and PI-ME/CFS participants showed force-related brain activation in the left M1, right cerebellum, and left putamen during the task. We next assessed group differences with t-test (at $p = 0.01$, $k > 65$), but there

was no difference between the groups. We also assessed changes across blocks with a two-way ANOVA (2 groups × 4 blocks), which showed that blood oxygen level dependent (BOLD) signal of PI-ME/CFS participants decreased across blocks bilaterally in temporo-parietal junction (TPJ) and superior parietal lobule, and right temporal gyrus in contradistinction to the increase observed in HVs (F (3,45) = 5.4, voxel threshold $p \leq 0.01$, corrected for multiple

**Fig. 1 | Patient recruitment and characteristics. a** Diagram showing the procedure followed to recruit adjudicated Post-infectious Myalgic Encephalomyelitis/ Chronic Fatigue Syndrome (PI-ME/CFS) and matched healthy volunteers (HV) and measurements undertaken for deep phenotyping of the cohorts. The number of participants at each stage of recruitment are noted. **b** Comparisons of age, sex, and BMI distribution in HV (blue; *n* = 21 independent participants) and PI-ME/CFS (red; *n* = 17 independent participants) using unadjusted two-sided t-tests for independent samples. **c** Distribution of the response of HV (blue; *n* = 21 independent participants) and PI-ME/CFS (red; *n* = 17 independent participants) to the indicated patient reported outcome questionnaires. Group comparisons performed using unadjusted two-sided Mann–Whitney-U tests. CSF cerebrospinal fluid, PBMC peripheral blood mononuclear cell, BMI body mass index, PROMIS Patient-Reported Outcomes Measurement Information System, PHQ-15 Patient Health Questionnaire – 15, MASQ Multiple Ability Self-Report Questionnaire. Figure 1A created with Biorender.com. Source data are provided as a Source Data file.

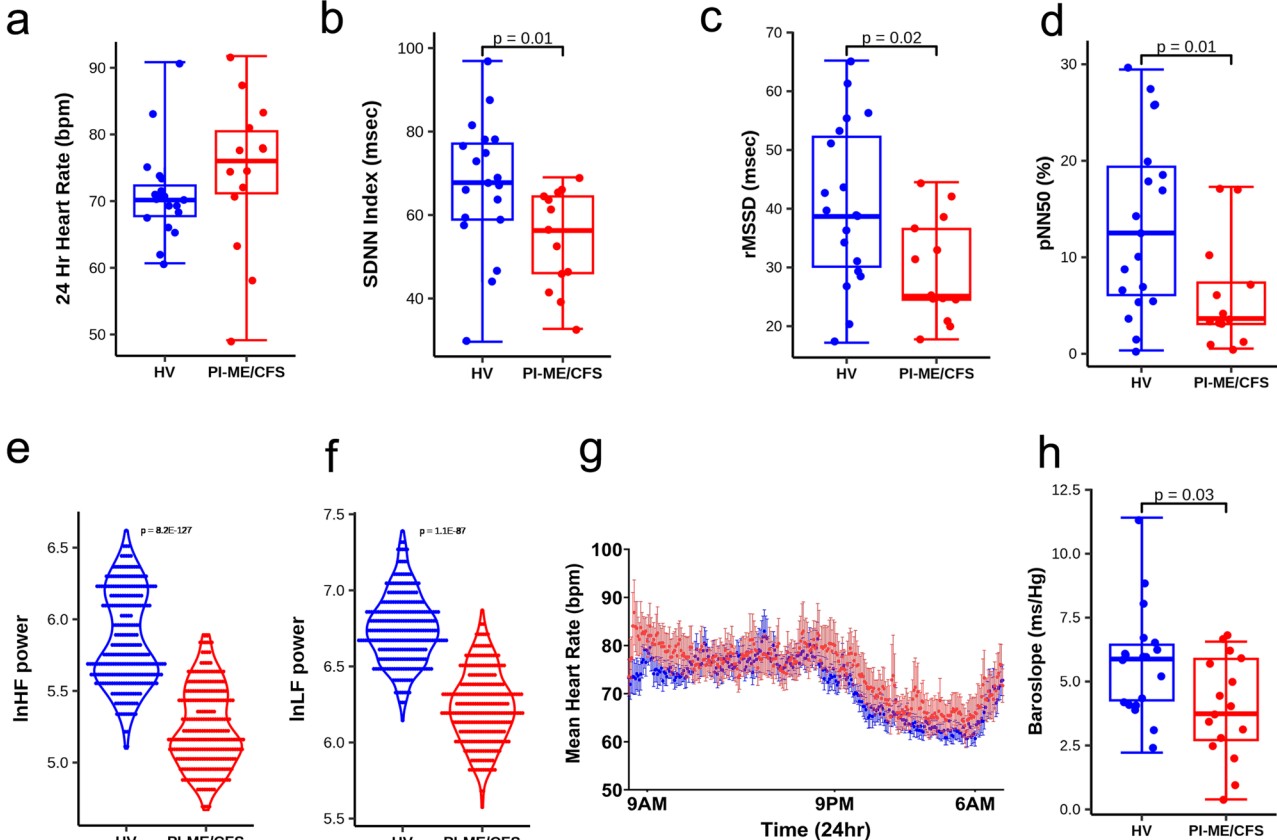

**Fig. 2 | Diminished heart rate variability measures are consistent with decreased parasympathetic activity in the PI-ME/CFS cohort compared to HV. a** Table of time and frequency domain heart rate variability measurements. Group comparisons for panel **a** were performed with unadjusted two-sided Mann–Whitney U tests. Box plots comparing HV (blue; *n* = 19 independent participants) and PI-ME/CFS (red; *n* = 14 independent participants) for (**b**) SDNNI (msec) (**c**) rMSSD (msec, *p* = 0.019, unadjusted two-sided t-test for independent samples with equal variance) (**d**) pNN50 (%, *p* = 0.017, unadjusted two-sided Mann–Whitney U test) (**e**) lnHF(ms²) (**f**) lnLF (ms²). Box plots depict the median (horizontal line) within quartiles 1–3 (bounds of box). Whiskers extend to minimum and maximum values **g**: Mean heart rate of HV (blue; *n* = 20 independent participants) and PI-ME/ CFS (red; *n* = 13 independent participants) of 5-min segmented intervals over a 24-h period graphed over 24-h period. Error bars represent ±SE for each 5-min time block for each group. Note HV graph (blue) demonstrates fluctuations throughout the day with subject heart rates displaced slightly higher, suggesting increased sympathetic activity. Similarly, the typical sinusoidal drop in heart rate over sleeping hours is diminished in subjects also suggesting diminished parasympathetic and/or increased sympathetic activity. **h** Box plot of baroreflex-cardiovagal gain as measured by mean baroslope (ms/mmHg). HVs (blue; *n* = 19 independent participants) and PI-ME/CFS (red; *n* = 16 independent participants) are compared using an unadjusted two-sided t-test for independent samples with equal variance (*p* = 0.015). Box plot H depicts the median (horizontal line) within quartiles 1–3 (bounds of box). Whiskers extend to minimum and maximum values. SDNNi standard deviation of the average NN intervals for each 5 min segment of a 24 h HRV recording, rMSSD root mean square of successive differences between normal heartbeats, pNN50 proportion of NN50 divided by the total number of NN (R-R) intervals, HF high frequency, LF low frequency, SD1 standard deviation of Poincaré plot of RR intervals perpendicular to the line-of-identity, SD2 standard deviation of the Poincaré plot of RR intervals along the line-of-identity. Source data are provided as a Source Data file.

comparisons *p* ≤ 0.05, *k* > 65; Fig. 4d, e). TPJ activity is inversely correlated with the match between willed action and the produced movement.

## Differential cardiorespiratory performance in PI-ME/CFS

Cardiopulmonary exercise testing (CPET) was performed on eight PI-ME/CFS and nine HVs (Supplementary Fig. S8A). During testing, all but one PI-ME/CFS participant reached the peak respiratory exchange ratio (RER) of ≥1.1 and there was no difference between the groups. Effort preference did not correlate with peak power in PI-ME/CFS participants (Supplementary Fig. S8B). The ratio of Ventilation/VCO₂, oxygen saturation levels in the quadricep muscle as measured by Near Infrared Spectroscopy, and gross mechanical efficiency were not different between groups (Supplementary Fig. S8C–E). These results suggest that PI-ME/CFS participants performed to the best of their abilities.

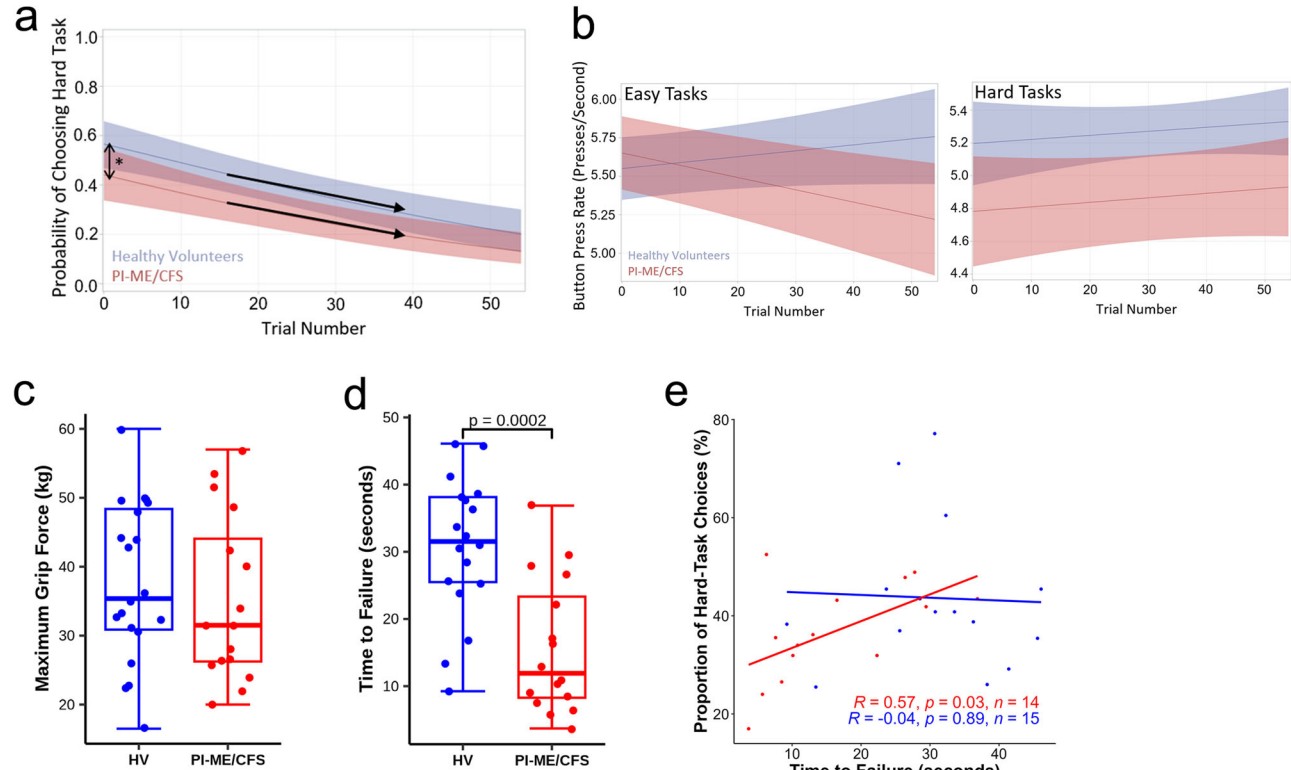

**Fig. 3 | Impaired effort measures and motor performance were observed in PI-ME/CFS cohort compared to HV. a, b** Effort-Expenditure for Rewards Task: **a** Probability of choosing the hard task is significantly more in HV (blue; $n = 16$ independent participants) compared with PI-ME/CFS (red; $n = 15$ independent participants) at the start of and throughout the task. The Odds Ratio for the probability of choosing the hard task at the start of the task is 1.65 [1.03, 2.65], $p = 0.04$ using Fisher's Exact test. The lines are the curvilinear fits and the error bands are the confidence intervals. Decline rates (i.e. response to fatigue) between the groups is similar as the trial progresses. **b** Button press rates for easy (right) and hard (left) tasks as the trial progresses is shown for HV (blue) and PI-ME/CFS (red) participants. The lines are the linear fits and the shaded error bands are the confidence intervals. The decline in button press rate over time during the easy tasks in PI-ME/CFS did not impact easy task performance, which is supportive of pacing in PI-ME/CFS participants. **c**–**e** Grip Strength test: Box plots of (**c**) maximum grip force of HV (blue; $n = 20$ independent participants) and PI-ME/CFS (red; $n = 16$ independent participants) and (**d**) time to failure of HV (blue; $n = 18$ independent participants) and PI-ME/CFS (red; $n = 16$ independent participants), unadjusted two-sided t-test for independent samples with equal variance, $p = 0.0002$. Correlation between time to failure and (**e**) proportion of hard task choices in HV (blue; $n = 15$ independent participants) and PI-ME/CFS (red; $n = 14$ independent participants). For figure **e**, the relationship between indicated variables in x and y axis were fitted by linear regression in each group. Linear regression t-tests were used to determine non-zero slope. Exact $p$ values of the correlations are presented on the graph. For box plots **c** and **d**, boxes depict the median (horizontal line) within quartiles 1–3 (bounds of box). Whiskers extend to minimum and maximum values. Source data are provided as a Source Data file.

Peak power ($p = 0.08$), peak respiratory rate, peak heart rate ($p = 0.07$), and peak VO2 ($p = 0.004$) were all lower in the PI-ME/CFS participants (Fig. 5a–d), a difference of approximately 3.3 metabolic equivalent of task units (METs). These peak measures did not correlate with effort preference in PI-ME/CFS. PI-ME/CFS participants performed at a lower percentage of their predicted $VO_{2peak}$ (Fig. 5e) as determined by the Wasserman-Hansen cycling equation[21,22] and at a lower percentage of their age-predicted maximal heart rate (Supplementary Fig. S8F). The lower peak heart rate and higher resting heart rate for PI-ME/CFS participants led to a lower heart rate reserve (Fig. 5f). Chronotropic incompetence, as measured by age-predicted HRR (%pHRR), was noted in five of eight PI-ME/CFS and one of nine HVs ($X^2$ (1, $n = 17$) = 4.9, $p = 0.03$). The slope of the heart rate response during the CPET was also lower in PI-ME/CFS participants compared to HVs ($1.03 \pm 0.16$ versus $0.70 \pm 0.27$, $p < 0.01$; Fig. 5g). Anaerobic threshold (AT) was achieved within a similar time period and occurred at lower VO2 levels in PI-ME/CFS (Fig. 5h), a difference of approximately 1.4 METs. $VO_2$ at AT also correlated with Type2:1 mFd (Supplementary Fig. S8J) in PI-ME/CFS but not HVs, evidence of concomitant muscular deconditioning. Salivary cortisol measurements taken at rest in the morning, at noon, and before sleep showed no group differences but were significantly lower in PI-ME/CFS participants one hour after CPET (Supplementary Fig. S8L). Thus, despite successful CPET engagement,

PI-ME/CFS participants were less likely to achieve their predicted maximal output, suggesting a differential cardiorespiratory performance related to autonomic function, hypothalamic-pituitary-adrenal axis hyporesponsiveness, and muscular deconditioning from disuse that clinically impacts activities of daily life[23]. This was supported by at-home waist actigraphy measures, demonstrating a lower step count and total activity for PI-ME/CFS participants due to less moderate intensity activity ($40.64 \pm 37.4$ versus $6.4 \pm 7.0$ min/day; $p = 0.007$).

Despite altered cardiorespiratory function, there were no clinically meaningful differences in dietary energy intake (Supplementary Data S11) or total body energy use, sleeping energy use, or respiratory quotient between the groups before or after CPET testing (Supplementary Fig. S9A–D; Supplementary Data S12).

## Increased cognitive symptoms but normal neurocognitive testing in PI-ME/CFS

PI-ME/CFS participants had more self-reported total cognitive complaints ($66.9 \pm 14.7$ versus $92.1 \pm 17.6$, $p = 0.00006$) and in all five cognitive domains measured: attention, verbal memory, visuoperceptual, language, and visual memory (Fig. 1c). In contrast, there were no group differences in performance of any of the 15 neuropsychological tests administered (Supplementary Data S13) or differential degradation of performance over time (Supplementary Information, Cognitive

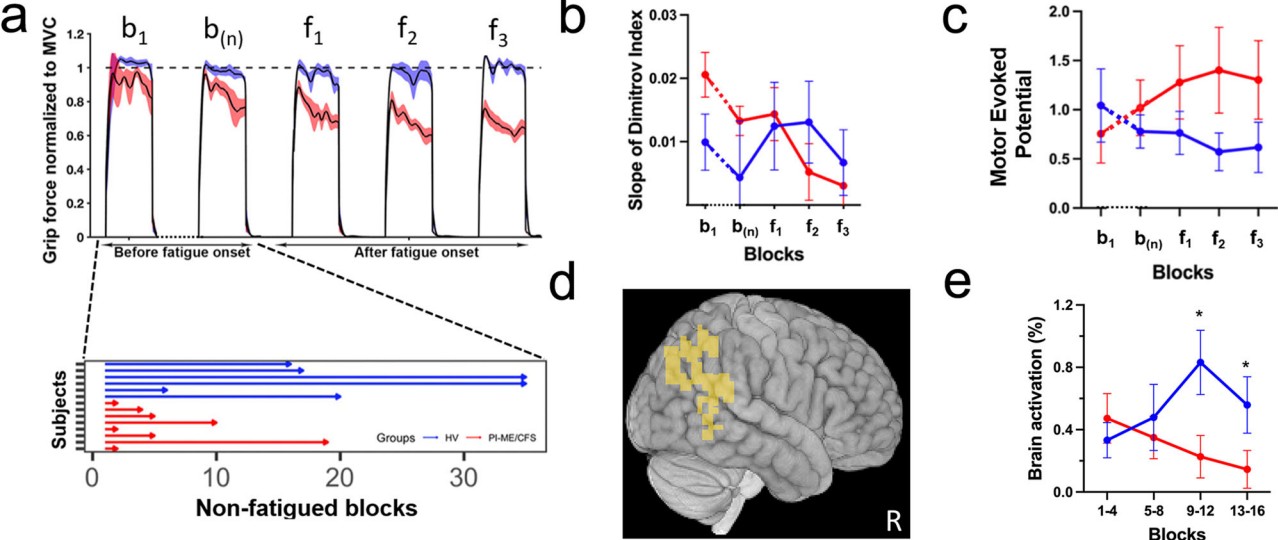

**Fig. 4 | Impaired sustained effort and motor performance was observed in PI-ME/CFS cohort compared to HV. a–e** Repetitive Grip Strength test: **a** Grip force normalized to maximum voluntary contraction (MVC) in the first block, the last block prior to fatigue onset, and the first three blocks after fatigue onset in HV (blue) and PI-ME/CFS cohorts (red). A significant grip force difference was noted between the groups (−1.2 ± 4 versus −6.4 ± 4 kilogram-force units, t(12) = 2.46, $p = 0.03$). The number of non-fatigued grip test blocks of HV (blue) and PI-ME/CFS (red) participants is also displayed. **b** Slope of the Dimitrov index across the first block ($b_1$), the last block prior to fatigue onset ($b_n$), and the first three blocks after fatigue onset ($f_1$, $f_2$, and $f_3$) in HV (blue; $n = 6$ independent participants) and PI-ME/CFS (red; $n = 8$ independent participants) patients. A significant difference was noted between the groups (0.2 ± 0.5 versus −0.43 ± 0.3, t(12) = 3.2, $p = 0.008$). **c** Mean and standard error of the motor evoked potential of HV (blue; $n = 6$

independent participants) and PI-ME/CFS (red; $n = 8$ independent participants) participants spanning the last five grip test blocks prior to fatigue onset. The amplitudes of the MEPs of HVs significantly decreased over the course of the task while the amplitudes of the MEPs of PI-ME/CFS participants significantly increased (−0.13 ± 0.2 versus 0.13 ± 0.2 MEP units; t(12) = 2.4, $p = 0.03$ D. Brain regions where Blood Oxygen Dependent (BOLD) signal decreased over grip strength blocks in PI-ME/CFS patients and increased over grip strength blocks in HVs. **e** Brain activation of the regions depicted in **d** measured in the blocks of four over the course of the experiment in HV (blue; $n = 10$ independent participants) and PI-ME/CFS (red; $n = 8$ independent participants) cohorts. For **e**, a two-way ANOVA was run where F(3,45) = 5.4 with a voxel threshold of $p \le 0.01$, corrected for multiple comparisons $p \le 0.05$, $k > 65$ ($p$-values were 0.976, 0.43, 0.02 (*), and 0.02 (*) for blocks 1 to 4 respectively). Source data are provided as a Source Data file.

Function). No correlations were noted between any of the 15 neuropsychological tests administered and effort preference.

## Differential cerebrospinal fluid catechols and metabolite profile in PI-ME/CFS

In cerebrospinal fluid, the PI-ME/CFS group had statistically significant decreased levels of DOPA, DOPAC, and DHPG (Fig. 6a–c). Levels of norepinephrine, cys-DOPA, and dopamine did not differ between the groups. These results did not change after excluding data from participants taking central-acting medications.

Additional analysis revealed that norepinephrine correlated with both time to failure and effort preference in PI-ME/CFS participants (Fig. 6d, e). Dopamine correlated with time to failure (Fig. 6f) but not with effort preference (Supplementary Fig. S11B). Among HVs, DHPG correlated with effort preference (Fig. 6g) but not with time to failure (Supplementary Fig. S11C). Several cognitive symptoms correlated with catechols in PI-ME/CFS participants (Supplementary Fig. S12A–I). In contradistinction, objective neurocognitive performance showed that only the Wisconsin Card Sort Test correlated with DOPA in PI-ME/CFS participants (Supplementary Fig. S12J). For HVs, several correlations between catechols and neuropsychological tests were observed (Supplementary Fig. S12K–O). These correlations suggest that perception and behavior, but not cognitive performance, are related to catechol levels in PI-ME/CFS participants.

Metabolomic analysis of cerebrospinal fluid also showed group differences (Fig. 6h). Tryptophan metabolites were among the top 15 differentially expressed and statistically significant after correction for multiple comparisons (Fig. 6i). Decreased glutamate, dopamine 3-O-sulfate, butyrate, polyamine, and tricarboxylic acid (TCA) pathway metabolites were noted in PI-ME/CFS participants (Supplementary Data S14A). Threonine and glutamine were decreased in males (Fig. 6j,

Supplementary Fig. S13D, Supplementary Data S14C). Several tryptophan metabolites were decreased in females suggesting a decrease in serotonin signaling (Fig. 6k, Supplementary Fig. S13E, Supplementary Data S14D). This was irrespective of NSRI/SSRI use.

## Immune activation and sex-based differences in PI-ME/CFS

There were no group differences in percentage of CD4, CD8, and CD19 cells in both peripheral blood and cerebrospinal fluid (Supplementary Data S15A and Supplementary Fig. S15C). An increase in percentage of naïve and decrease in switched memory B-cells in blood were observed in PI-ME/CFS participants (Fig. 7a, b). Two PI-ME/CFS participants had detectable antibody secreting B-cells in cerebrospinal fluid; these participants did not have oligoclonal bands or autoantibodies. There was no group difference in NK cell frequency, but a subset of PI-ME/CFS participants had an elevated percentage of NK CD56 bright/dim ratio in cerebrospinal fluid. No group differences were noted in NK lytic units, CD16/CD56 ratio, and plasma GDF-15 levels in blood. Discovery assays for a small panel of autoantibodies detected low levels in one PI-ME/CFS and two HVs.

Markers of T-cell activation, PD-1$^+$ CD8 T-cells, were elevated in the cerebrospinal fluid of PI-ME/CFS participants (Fig. 7c). PD-1$^+$ CD4 T-cells did not change in blood or cerebrospinal fluid. To expand on these observations, other T-cell markers (TIGIT, CD244, and CD226) were analyzed in a subset of participants. CD226$^+$ CD8 T-cells were decreased in blood (Fig. 7d) of PI-ME/CFS participants with no change in expression of CD244 or TIGIT (Supplementary Data S15B, D).

When stratified by sex, PI-ME/CFS males had increased CXCR5 expression on CD8 + T-cells in cerebrospinal fluid (Fig. 7e). PI-ME/CFS females had increased CD8+ naïve T-cells in blood (Fig. 7f), indicating different subpopulation of immune cells are enriched in the male and female PI-ME/CFS groups.

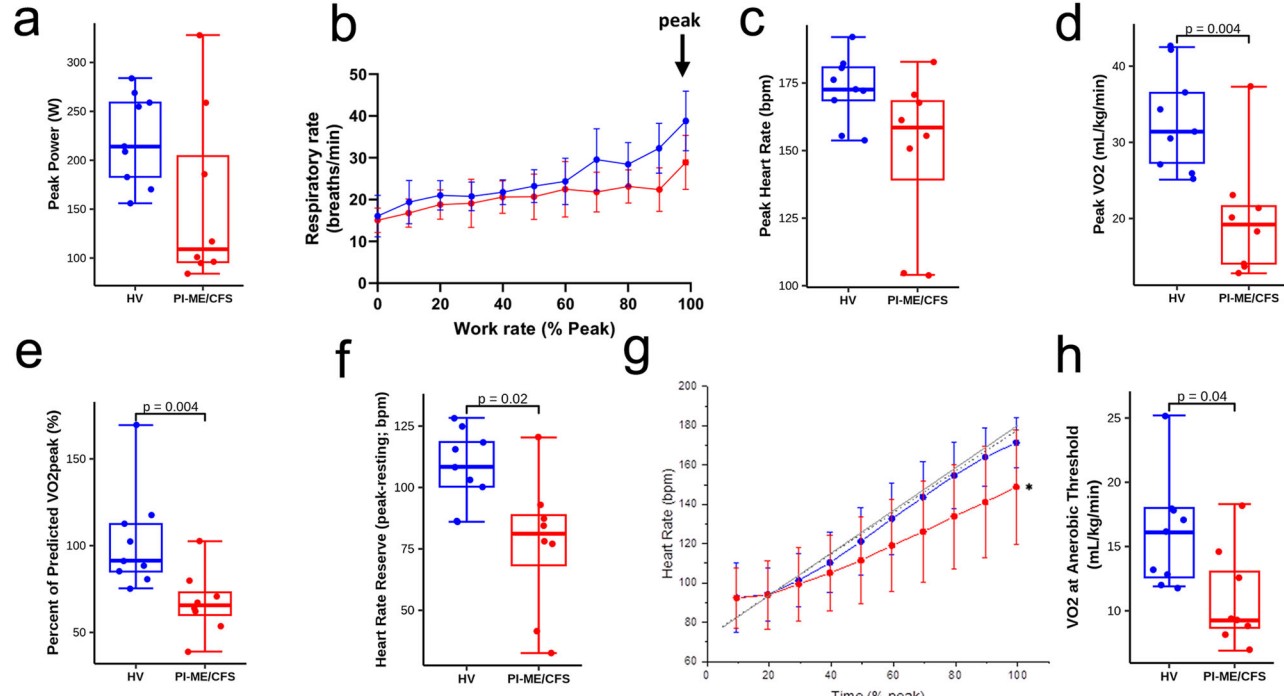

**Fig. 5 | Impaired cardiopulmonary performance was observed in PI-ME/CFS cohort compared to HV.** Cardiopulmonary Exercise Test (CPET) **a**–**h**: Box plots of (**a**) peak power attained during CPET for HV (blue; *n* = 9 independent participants) and PI-ME/CFS (red; *n* = 8 independent participants) participants, (**b**) peak respiratory rate with the error bars representing ±SD at each work rate, (**c**) peak heart rate (unadjusted two-sided t-test for independent samples with unequal variance, *p* = 0.072), (**d**) Peak VO2 (unadjusted two-sided t-test with equal variance, *p* = 0.002), (**e**) Percent of predicted VO2 achieved of HV (blue; *n* = 9 independent participants) and PI-ME/CFS (red; *n* = 8 independent participants) participants (unadjusted two-sided t-test for independent samples with equal variance, *p* = 0.004), and (**f**) heart rate reserve for HV (blue; *n* = 9 independent participants) and PI-ME/CFS (red; *n* = 8 independent participants) participants (unadjusted two-sided t-test for independent samples with equal variance, *p* = 0.011). **g** Heart rate as a function of % total CPET time depicted as expected for age and gender of HV (*n* = 9 independent participants) and PI-ME/CFS (*n* = 8 independent participants) (gray solid and dashed lines, respectively, unadjusted two-sided Mann–Whitney U test for independent samples). Mean heart rate responses from the CPET are depicted for HV (blue line) and PI-ME/CFS (red line). A significant difference was observed for the heart rate slope between PI-ME/CFS and expected (0.70 ± 0.27 versus 1.05 ± 0.12, *p* = 0.014), but not for HV and expected (1.03 ± 0.16 versus 1.08 ± 0.13; *p* = 0.479). The deviation from expected relation reflects chronotropic incompetence in the PI-ME/CFS group. **h** Box plot of VO2 at the anaerobic threshold (AT) in HV (blue; *n* = 9 independent participants) and PI-ME/CFS (red; *n* = 8 independent participants) using an unadjusted two-sided t-test for independent samples with equal variance (*p* = 0.024). For box plots **a**, **c**–**f**, and **h** boxes depict the median (horizontal line) within quartiles 1–3 (bounds of box). Whiskers extend to minimum and maximum values. Source data are provided as a Source Data file.

Analysis of PBMC gene expression for all participants by principal component analysis (PCA) revealed no outliers (Fig. 8a). 614 differentially expressed (DE) genes clustered the samples based on disease status. However, exploration of PCA plot showed that samples clustered based on their sex (Fig. 8b). Assessment of interaction between sex and disease status was not significant. This suggests sex is a potential confounder impacting gene expression. Hence DE analysis in male and female cohorts was performed, which identified distinct subsets of DE genes, with only 34 (<5%) genes overlapping between them (Fig. 8c, Supplementary Data S16A, B). PCA of DE genes in each sex cohort clustered the samples into distinct HV and PI-ME/CFS groups (Fig. 8d, f).

887 DE genes in male HV and PI-ME/CFS cohorts enriched (*p* < 0.05) in ubiquitin, IL-10, T-cell, and NF-kB pathways (Fig. 8h, i). DE genes related to the STAT4-TLR9 protein-protein interactome and were upregulated in male PI-ME/CFS participants (Supplementary Fig. S14A). 849 DE genes identified in female HV and PI-ME/CFS cohorts were enriched in B-cell proliferation processes (Fig. 8j, k) and DE genes identified in the B-cell interactome were upregulated in female PI-ME/CFS participants (Supplementary Fig. S14B). Additionally, DE genes were also enriched in cytokine and lymphocyte proliferation processes. These data are consistent with expansion of naïve B-cells by the STAT4-TLR9 and other B-cell pathways observed in PI-ME/CFS by flow cytometry.

PCA analysis of aptamers measured in serum and cerebrospinal fluid did not identify outliers. Univariate analysis of the aptamer datasets did not identify any FDR-corrected statistically significant differential features between PI-ME/CFS and HV participants (Supplementary Data S17A, B). Within each sex separately, exploratory multivariate analyses suggested a subset of aptamers were predictive of PI-ME/CFS status, which are reported for potential validation (Supplementary Fig. S15A–D, S16A–D; Supplementary Data S17C–F).

Sex differences in PI-ME/CFS were also assessed in a publicly deposited ME/CFS RNASeq dataset (GSE130353) from monocytes (Supplementary Fig. S17A–D). Of 638 DE genes in males and 420 DE genes in females, only 23 genes were common (Supplementary Fig. S17E). These observations support that distinct biological processes are perturbed in male and female PI-ME/CFS patients.

## Sex-based differences in muscle gene expression profile in PI-ME/CFS

PCA analysis of muscle gene expression did not identify any outliers (Fig. 9a). Univariate analysis did not identify any FDR-corrected statistically significant differential features (Supplementary Data S19A). The PCA identified clustering based on sex (Fig. 9b). Only 15 DE genes were common in male and female cohorts (Fig. 9c), consistent with prior observations. In the male PCA of 593 DE genes, clustering was observed based on the disease status (Fig. 9d, e). Genes upregulated in

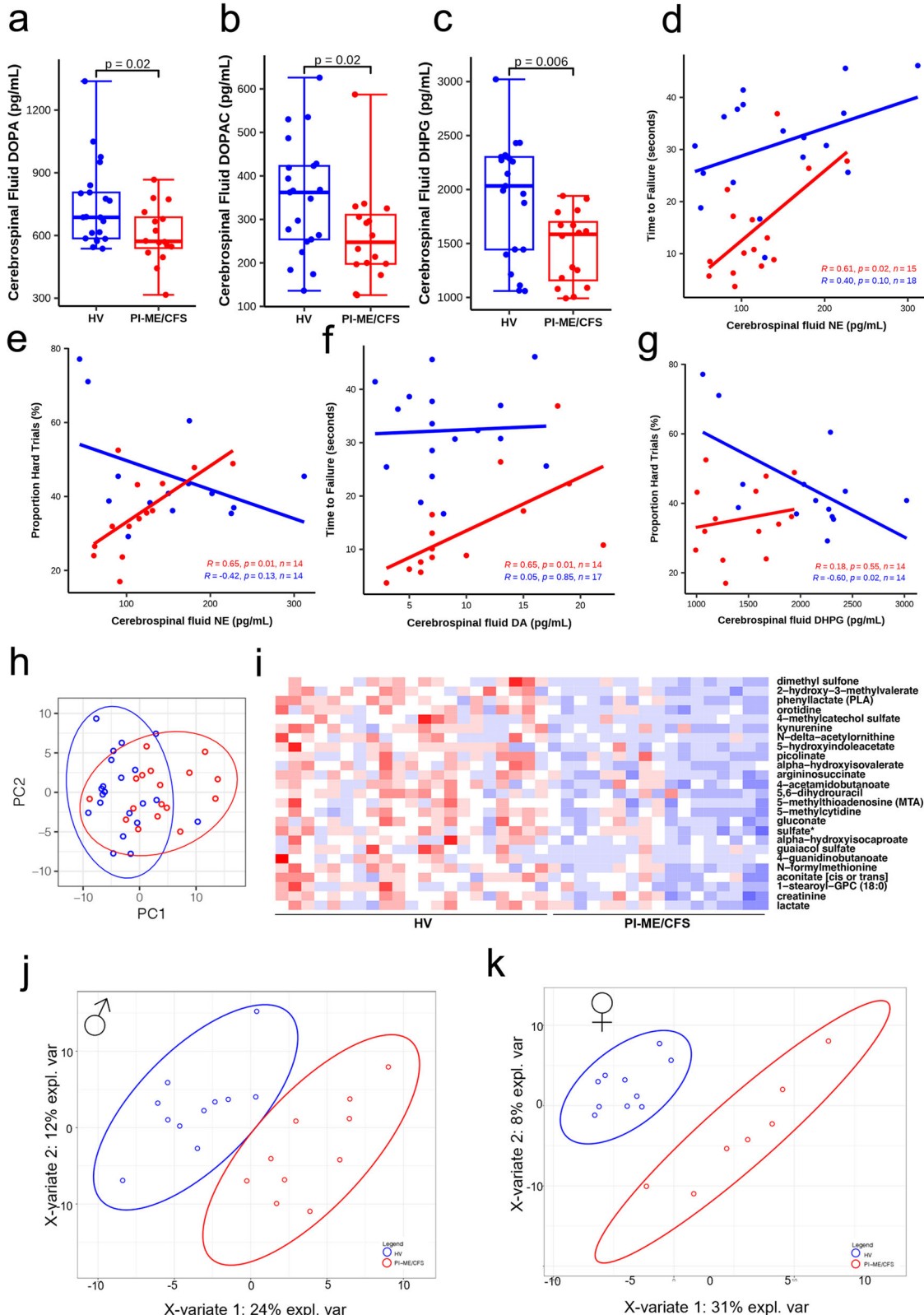

the PI-ME/CFS males were enriched in epigenetic changes and processing of RNA and the downregulated genes were enriched in hexose metabolism and mitochondrial processes (Fig. 9h–i). Several genes involved in fatty acid beta-oxidation were upregulated, in an interactive network (Supplementary Fig. S18A). In the females, PCA of the 328 DE genes showed distinct clusters of HV and PI-ME/CFS samples (Fig. 9f, g). DE genes upregulated in the PI-ME/CFS females were enriched in growth hormone receptor signaling and ubiquitin transferase function (Fig. 9j). Downregulated genes were involved in fatty acid oxidation and mitochondrial processes (Fig. 9k). A subset of these downregulated fatty acid oxidation genes was highly inter-connected (Supplementary Fig. S18B). Skeletal fatty acid oxidation is regulated during exercise as fatty acid uptake, is increased after moderate exercise[24,25], and is downregulated in muscle deconditioning[26]. These

**Fig. 6 | Differential catecholamines and tryptophan pathway metabolites levels in PI-ME/CFS patients cerebrospinal fluid. a–c** Box plot of the indicated neurotransmitters on y axis in HV (blue; $n = 21$ independent participants) and PI-ME/CFS (red; $n = 16$ independent participants) in (**a**) unadjusted two-sided Mann–Whitney U test ($p = 0.021$), (**b**), unadjusted two-sided Mann–Whitney U test ($p = 0.025$), and (**c**) unadjusted two-sided t-test for independent samples with equal variance ($p = 0.006$). For box plots **a–c** boxes depict the median (horizontal line) within quartiles 1–3 (bounds of box). Whiskers extend to minimum and maximum values. Correlation between cerebrospinal fluid norepinephrine (NE) and (**d**) time to failure on grip strength task in HV (blue; $n = 18$ independent participants) and PI-ME/CFS (red; $n = 15$ independent participants) or (**e**) proportion of hard task choices (i.e., effort preference) in HVs (blue; $n = 14$ independent participants) and PI-ME/CFS (red; $n = 14$ independent participants). **f** Correlation between cerebrospinal fluid dopamine and time to failure in HVs (blue; $n = 17$ independent participants) and PI-ME/CFS (red; $n = 14$ independent participants). **g** Correlation between cerebrospinal fluid DHPG and proportion of hard task choices (i.e., effort preference) in HVs (blue; $n = 14$ independent participants) and PI-ME/CFS (red; $n = 14$ independent participants). For figures **d–g**, the relationship between indicated variables in x and y axis were fitted by linear regression in each group. The exact $p$ value of each regression is presented on the graph, linear regression t-test for nonzero slope. **h** PCA computed from all metabolites measured from the cerebrospinal fluid samples in the indicated groups. **i** Heatmap of statistically significant (false discovery rate adjusted $p$-value < 0.05) differentially expressed metabolites in the indicated groups on x axis and the metabolites labeled on y axis. Red: upregulated; Blue: downregulated. Supervised clustering of metabolites measured from the cerebrospinal fluid samples in (**j**) male cohorts and (**k**) female cohorts from PLSDA analysis. DHPG (S)–3,5-Dihydroxyphenylglycine, PLSDA Partial least square discriminant analysis, PC principal component. Source data are provided as a Source Data file.

results suggest sex differential in the muscular bioenergetics and muscular deconditioning of PI-ME/CFS participants[27].

### Lack of differences in lipidomics between PI-ME/CFS and healthy volunteers

Univariate analysis of the plasma lipidomic data did not identify statistically significant differences between HV and PI-ME/CFS groups (Supplementary Data S20). Multivariate analysis in all participants as well as in male and female cohorts separately identified several lipids as important variables in prediction (Supplementary Fig. S19A–C) and are consistent with a prior lipidomic analysis[28].

### Differential fecal microbiota in PI-ME/CFS by shotgun metagenomics

Differences in the alpha diversity of stool samples (Supplementary Fig. S20A, B, D) were noted with HVs having a greater number of taxa within the samples (Number of Observed Features, $p = 0.002$) but no differences in average proportional abundance (Inverse Simpson Index, $p = 0.79$). Beta diversity, as measured by Bray–Curtis dissimilarity (Supplementary Fig. S20C), demonstrated significant differences in microbial community composition between the groups (PERMANOVA $p < 0.0081$).

### Validation of diagnosis and course of illness on longitudinal follow-up

Within four years of participation, four of the 17 PI-ME/CFS participants had a spontaneous full recovery. No other new medical diagnoses were reported by the PI-ME/CFS participants.

## Discussion

This study obtained a more extensive set of biological measurements in people with PI-ME/CFS than any previous study. Although the number of study subjects was small compared to the prior literature, it identified biological alterations and confirmed some previously reported biological alterations.

All patients had documentation of good health followed by an episode of infection that led to ME/CFS symptoms. All PI-ME/CFS participants reported clinically substantial fatigue, physical symptoms, and decreased functional status that was 1.5 to two standard deviations worse than the general population. Cases were reviewed by a panel of adjudicators that had to unanimously agree on the case validity to be included in this cohort. Only 10% of those with completed reviews were adjudicated as PI-ME/CFS cases. Even in the cases that met study criteria, diagnostic misattribution was noted in 20%, with underlying causes becoming manifest over time. This misclassification bias has important ramifications on the interpretation of the existing ME/CFS research literature[8].

Another challenge to the characterization of ME/CFS is concerns about the validity of the manifestations due to depression or anxiety[29].

Our cohort underwent substantial performance validity testing using neuropsychological measures and demonstrated consistency of response, suggesting that their symptoms were reliable and a true representation of their disease. Even though PI-ME/CFS participants endorsed more depressive and anxiety symptoms than HVs, they did not meet psychiatric diagnostic criteria. There was also no difference in psychiatric history or reporting of traumatic events between the two groups. Thus, psychiatric disorders were not a major feature in this cohort and did not account for the severity of their symptoms.

We first determined the physiological basis of fatigue in PI-ME/CFS participants. The notion of fatigue, as we use it, is a limit on ability or a diminution of ability to perform a task. Effort preference is how much effort a person subjectively wants to exert. It is often seen as a trade-off between the energy needed to do a task versus the reward for having tried to do it successfully. If there is developing fatigue, the effort will have to increase, and the effort:benefit ratio will increase, perhaps to the point where a person will prefer to lose a reward than to exert effort. Thus, as fatigue develops, failure can occur because of depletion of capacity or an unfavorable preference.

There were no differences in ventilatory function, muscle oxygenation, mechanical efficiency, resting energy expenditure, basal mitochondrial function of immune cells[30], muscle fiber composition, or body composition supporting the absence of a resting low-energy state. However, substantial differences were noted in PI-ME/CFS participants during physical tasks. Compared to HVs, PI-ME/CFS participants failed to maintain a moderate grip force even though there was no difference in maximum grip strength or arm muscle mass. This difference in performance correlated with decreased activity of the right temporal-parietal junction, a part of the brain that is focused on determining mismatch between willed action and resultant movement[31]. Mismatch relates to the degree of agency, i.e., the sense of control of the movement[32]. Greater activation in the HVs suggests that they are attending in detail to their slight failures, while the PI-ME/CFS participants are accomplishing what they are intending. This was further validated by measures of peripheral muscular fatigue and motor cortex fatigue[20,33] that increased only in the HVs. Thus, the fatigue of PI-ME/CFS participants is due to dysfunction of integrative brain regions that drive the motor cortex, the cause of which needs to be further explored. This is an observation not previously described in this population.

Additionally, the results suggest the impact of effort preference, operationalized by the decision to choose a harder task when decision-making is unsupervised and reward values are held constant, on performance. 32–74% of the variance in time to grip failure for the PI-ME/CFS participants correlated with effort preference, which was not seen in HVs. This was accompanied by reduced brain activation in the right temporal-parietal area in PI-ME/CFS participants. Interviews with PI-ME/CFS participants revealed that sustained effort led to post-exertional malaise[34]. Conscious and unconscious behavioral

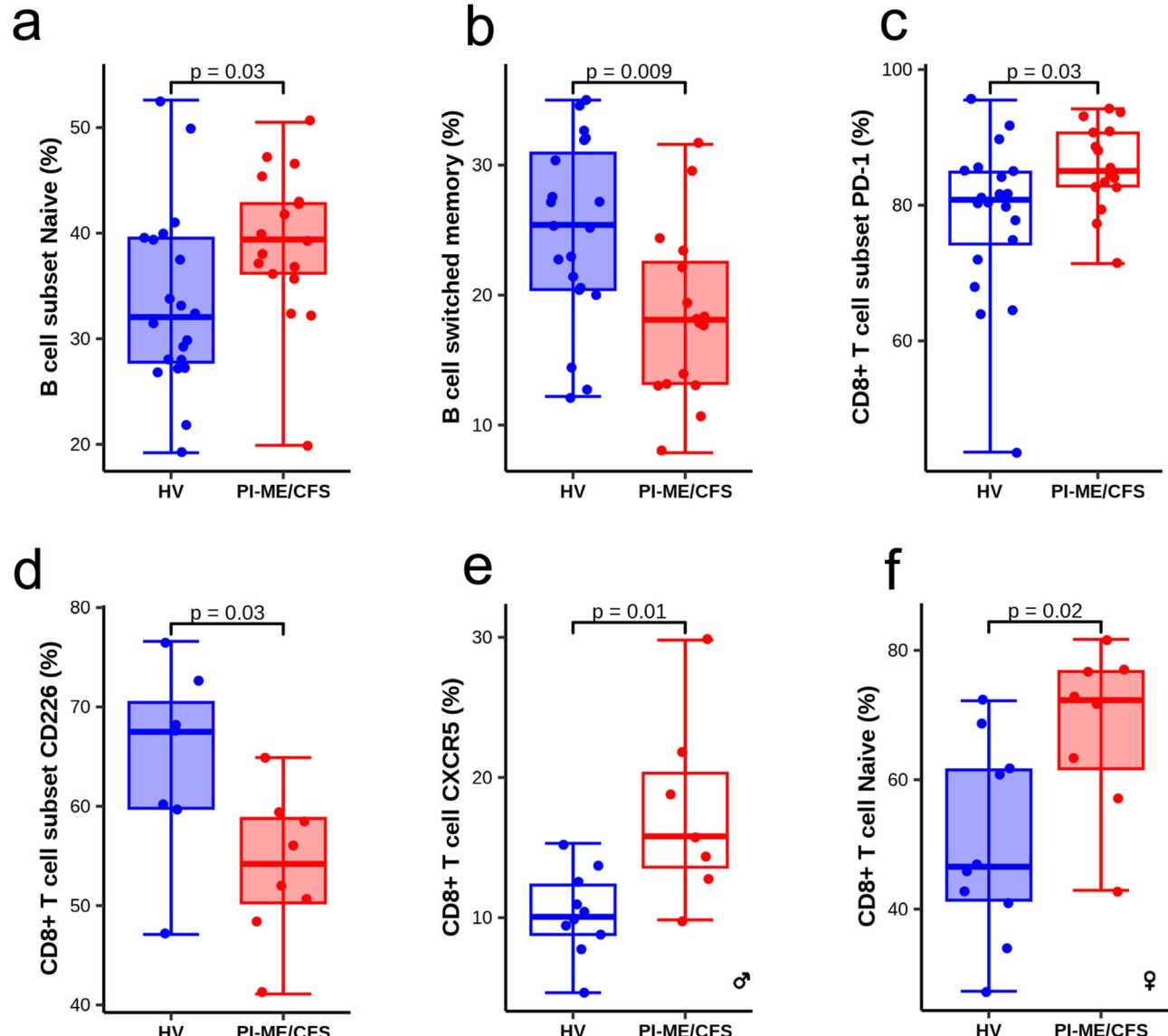

**Fig. 7 | Flow cytometry demonstrates distinct perturbation in immune cell subpopulation in PBMCs. a** Boxplot of the B cell subset naïve (%) in HV (blue; $n = 20$ independent participants) and PI-ME/CFS (red; $n = 17$ independent participants) groups, unadjusted two-sided t-test for independent samples with equal variance ($p = 0.037$). **b** Boxplot of B cell switched memory in HV (blue; $n = 20$ independent participants) and PI-ME/CFS (red; $n = 16$ independent participants) groups, unadjusted two-sided t-test for independent samples with equal variance ($p = 0.008$). **c** Boxplot of the CD8 + T cell subset PD-1 (%), Mann–Whitney U test, exact $p$-value = 0.033. **d** Boxplot of the CD8 + T cell subset CD226 (%), unadjusted two-sided t-test for independent samples with equal variance ($p = 0.055$). The samples used for boxplot (**d**) were collected at a separate time point than the others

boxplots; HV (blue; $n = 7$ independent participants) and PI-ME/CFS (red; $n = 8$ independent participants) groups. **e** Boxplot of CD8 + T cell CXCR5 (%), unadjusted two-sided t-test for independent samples with equal variance ($p = 0.014$). **f** Boxplot of CD8 + T cell naïve (%), unadjusted two-sided t-test for independent samples with equal variance ($p = 0.016$). Where indicated the plots shows the measurements from female and male cohorts. Measurements in PBMC samples are shown in shaded box and cerebrospinal fluid samples in open box. For box plots **a**–**f** boxes depict the median (horizontal line) within quartiles 1–3 (bounds of box). Whiskers extend to minimum and maximum values. Source data are provided as a Source Data file.

alterations to pace and avoid discomfort may underlie the differential performance observed[35].

We measured peripheral fatigue (high:low ratio) and central fatigue (post exercise depression). Both types of fatigue were seen in the HVs but not in the PI-ME/CFS participants. Moreover, testing of effort preference and the participants' own words (Supplementary Information, p.10) are consistent with this finding. Together these findings suggest that effort preference, not fatigue, is the defining motor behavior of this illness.

Consistent with this observation, with strong encouragement during CPET, all but one of the PI-ME/CFS participants reached a respiratory exchange ratio of 1.1. The PI-ME/CFS group showed a clear

reduction in cardiorespiratory fitness by reaching the AT at a lower level of work and ventilation[36]. At maximal performance, a substantial group difference in cardiorespiratory capacity became apparent, which was related to both chronotropic incompetence and physical deconditioning. These observations provide clarity to previous studies with inconsistent results[37]. Thus, incorporation of measures of autonomic tone and physical condition into studies of exercise performance is critical to understand the mechanisms of diminished cardiorespiratory function in ME/CFS[38].

A frequent complaint in our PI-ME/CFS cohort was cognitive dysfunction. This did not correlate with anxiety or depression measures. Standard clinical laboratory tests, brain imaging, measures of

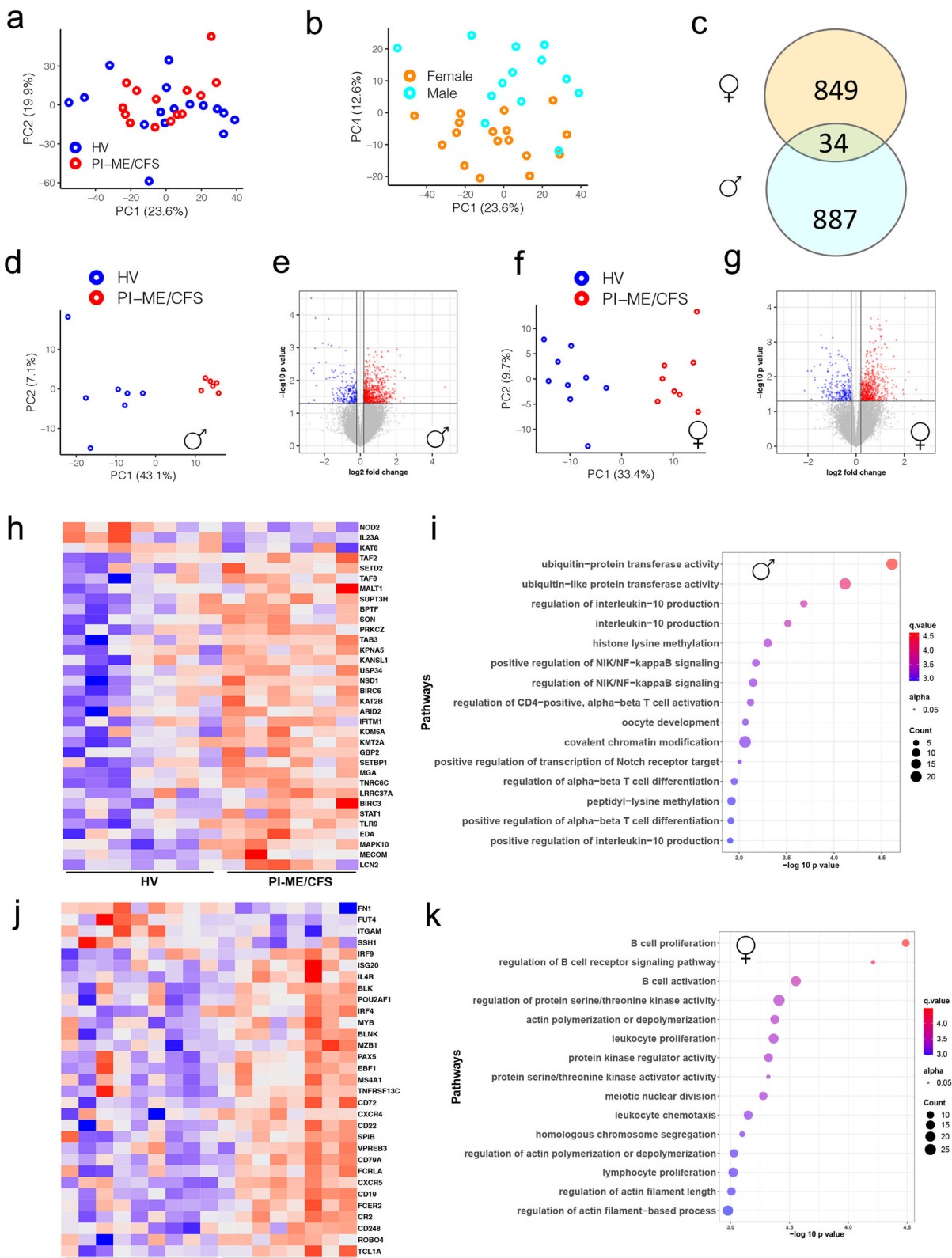

brain injury, and sleep architecture were unremarkable. Neuropsychological testing showed that even though the HV and PI-ME/CFS participants started with different levels of perceived mental and physical fatigue, there were no differences in cognitive performance. Further, both cognitive performance and perceived fatigue changed at the same rate during testing in both groups. This is consistent with the absence of a homogenous cognitive deficit in ME/CFS. Previous studies

suggest a small, heterogenous deficit in this population[39] which may not be evident in our study due to the small sample size. Performance validity testing is not routinely performed in the literature; inclusion of invalid performances would also bias studies toward differential performance.

Taken together, this evidence suggests that physical and cognitive fatigue may be mechanistically different. Interestingly, PI-ME/CFS

**Fig. 8 | Male and female cohorts have distinct perturbation in immune cell subpopulation and biological processes in PBMCs. a**, **b** PCA computed from all gene expression values with indicated groups: HV (blue), PI-ME/CFS (red) clusters or males (turquoise) and females (orange) highlighted for the indicated PCs. **c** Venn diagram showing common DE genes identified from male and female cohorts (*p* value < 0.05 as filter for DE genes using an unadjusted two-sided moderated t-test). DE analysis on the (**d**, **e**, **h**, **i**) male cohorts and (**f**, **g**, **j**, **k**) female cohorts. **d**, **f** PCA plots computed from DE genes shows robust clustering of samples based on the PI-ME/CFS status. **e**, **g** Volcano plots shows log transformed statistically significant (unadjusted *p*-values < 0.05 using an unadjusted two-sided moderated t-test) up (red) and down regulated (blue) genes in PI-ME/CFS male and female cohorts,

respectively. **h**, **j** Heatmaps of a subset of T cell process genes in males and B cell related processes in females. **i**, **k** Pathway enrichment plots showing top 15 pathways for which the DE genes from male and female cohorts selected for. The top 15 pathways are labeled on the y axis and the color of the circle is scaled with −log-10 *p*-value and the size of the pathway circles inside the plot are proportional to the number of genes that overlapped with the indicated pathway. Fisher's exact test was used in the *'clusterProfiler'* R package to obtain the log-transformed p-values. Red color nodes: upregulated in PI-ME/CFS and blue color nodes: downregulated in PI-ME/CFS. DE differentially expressed. Source data are provided as a Source Data file.

participants' catechol levels in cerebrospinal fluid correlated with grip strength and effort preference, and several metabolites of the dopamine pathway correlated with several cognitive symptoms. This suggests that central nervous system catechol pathways are dysregulated in PI-ME/CFS and may play a role in effort preference and cognitive complaints[40]. The pattern suggests decreased central catecholamine biosynthesis in PI-ME/CFS. Similarly, decreased serum catechols and their metabolites have recently been reported in Long COVID-19;[41,42] testing of cerebrospinal fluid has not yet been reported. Autonomic testing revealed HRV measures consistent with an increase in sympathetic and a decrease in parasympathetic activity in PI-ME/CFS, and a decreased baroslope and prolonged pressure recovery after Valsalva consistent with previous observations of autonomic dysfunction in ME/CFS[6,43].

We investigated several additional biological functions in PI-ME/CFS. Previous studies have implicated abnormalities in immunity, mitochondrial function, reduction-oxidation regulation, or altered microbiome structure in this condition[1,3,44,45]. There were increased naïve B-cells and decreased switched memory B-cells in blood of PI-ME/CFS participants. However, contrary to prior published work[46], there was no consistent pattern of autoimmunity across all PI-ME/CFS participants and autoantibody discovery assays did not reveal previously undescribed antibodies. PD-1, a marker of T-cell exhaustion and activation, was elevated in the cerebrospinal fluid of PI-ME/CFS participants. Although NK cell function was not different between groups in blood, they showed decreased expression of a cytolytic function marker in the spinal fluid. Previous studies suggest that NK cell function is decreased in ME/CFS[47–51], which may not be evident in our study due to the small sample size.

Autonomic measures revealed an increase in sympathetic and a decrease parasympathetic activity in PI-ME/CFS that cannot be attributed to depression or anxiety, which is consistent with previous observations[6,43]. There was also distinct gene expression, immune cell populations, metabolite, and protein profiles in male and female PI-ME/CFS participants. In the male PI-ME/CFS cohort, PBMC gene expression profiling showed perturbations in the T-cell activation, proteasome and NF-kB pathways. Analyses of serum and cerebrospinal fluid proteins identified several molecules related to innate immunity suggesting a distinct expression pattern in male cohorts. In contrast, in the female cohorts, gene expression profiling of PBMCs identified perturbations in B-cell and leukocyte proliferation processes with a corresponding identification of plasma lymphotoxin α1β2, which may act as a proliferative signal in secondary lymphoid tissues. Additionally, elevated levels of plasminogen in serum and eosinophil protein galactin-10, the C-C motif chemokine (MDC), and the IL 18 receptor accessory protein were present in cerebrospinal fluid of female PI-ME/CFS participants. Plasminogen and lymphotoxin α1β2 are known to activate proinflammatory states via NF-kB[52,53]. Due to the small sample size of our study, we further confirmed sex differences in a previously published data set[54]. Notably, only 2% of the differentially expressed elements overlapped between the sex-separated cohort. This suggests that different immunological phenotypes may distinguish the PI-ME/CFS phenotype based on sex. The cause of immune dysregulation is

not clear but may suggest the possibility of persistent antigenic stimulation[55]. Analysis of the gut microbiome showed differences in alpha and beta diversity consistent with previous studies[3,56]. Their potential role in modulating the immune profile needs further exploration.

Sex-driven group differences were also observed in gene expression profiles of muscle. Male PI-ME/CFS participants had upregulation of fatty acid beta-oxidation genes and down regulation of TRAF and MAP-kinase regulated genes. The female PI-ME/CFS participants had downregulation of fatty acid metabolism and mitochondrial processes in muscle. Thus, both males and females PI-ME/CFS samples had increased oxidative stress, but distinct pathways were perturbed in both groups.

Metabolomics of cerebrospinal fluid identified downregulation of tryptophan metabolites in the PI-ME/CFS cohort, consistent with prior ME/CFS and Long COVID-19 studies[41,57–59]. In female cohorts, tryptophan, butyrate, and tricarboxylic acid related compounds were identified as important variables in classifying HV and PI-ME/CFS groups. In male cohorts, a different subset of metabolites was identified as important classification predictors. These findings need to be validated in larger PI-ME/CFS cohorts.

This study has several limitations. This was a cross-sectional, exploratory case-control study. These data assess correlation, not causality. The sample size of the cohort was small but was very carefully characterized and was adequate for capturing group differences, noting important negative findings, and looking for patterns of consilience between measures. Consilience was noted among the multiple autonomic measures suggestive of decreased parasympathetic activity in PI-ME/CFS, among flow cytometry and RNA sequencing measures of adaptive immunity, and in the impact of sexual dimorphism across immunologic and metabolomic measurements. The small sample size may impact the precision of effect size measures, with a propensity to inflate estimates[60]. Post-hoc calculations of the effect size for a phenotyping sample of 21 and 17 participants to achieve a power of 80% is 0.94, suggesting only large effects will be noted to be statistically significant. Overall, the groups are well-matched on demographic factors, but individual HV and PI-ME/CFS participants were not always perfectly matched. Even though a major effort was made to minimize misattribution, we cannot completely exclude this possibility.

While we have focused on the positive results above, there were many results that were not different between HV and PI-ME/CFS participants (Supplementary Data S22). While it is possible that small differences in the groups may be demonstrable in larger cohorts, these negative results demonstrate that these findings are not required for the PI-ME/CFS phenotype and are poor targets for clinical intervention.

Considering all the data together, PI-ME/CFS appears to be a centrally mediated disorder. We posit this hypothetical mechanism of how an infection can create a cascade of physiological alterations that lead to the PI-ME/CFS phenotype (Fig. 10). Exposure to an infection leads to concomitant immune dysfunction and changes in microbial composition. Immune dysfunction may be related to both innate and adaptive immune responses that are sex dependent. One possibility is

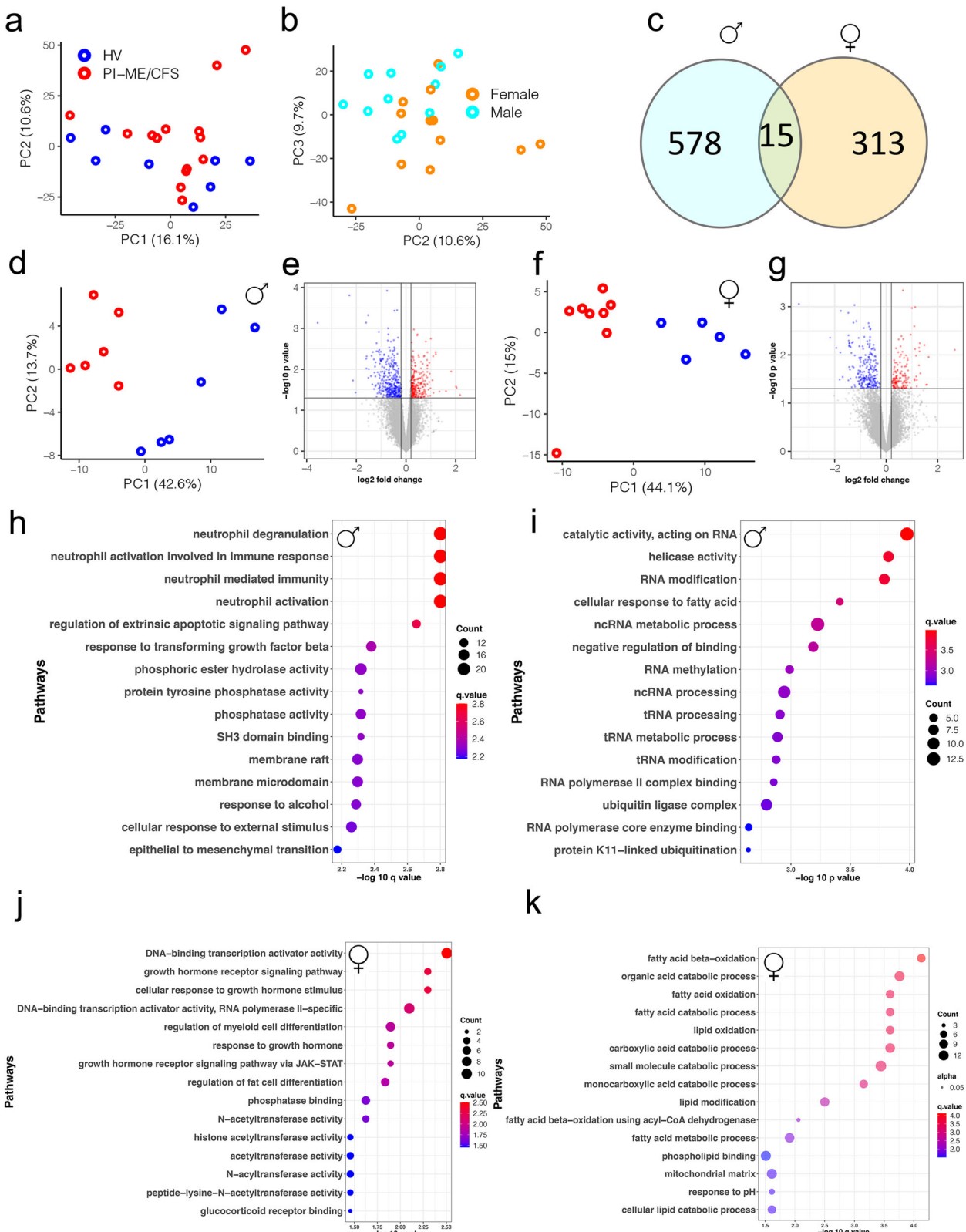

that these changes are related to antigen persistence of the infectious pathogen[61–63].

These immune and microbial alterations impact the central nervous system, leading to decreased concentrations of metabolites, including glutamate, tryptophan, spermidine, citrate, and the metabolites of dopamine (DOPAC) and norepinephrine (DHPG). The altered biochemical milieu impacts the function of brain structures. The catecholamine nuclei release lower levels of catechols, which impacts the autonomic nervous system leading to decreased heart rate variability and decreased baroreflex cardiovascular function, with downstream effects on cardiopulmonary capacity. Concomitant alteration of hypothalamic function leads to decreased activation of the temporoparietal junction during motor tasks, leading to a failure of the integrative brain regions necessary to drive the motor cortex. This

**Fig. 9 | Male and female cohorts have distinct differential gene expression profiles in the muscle. a**, **b** PCA computed from all gene expression values in samples highlighted from the indicated groups: HV (blue) and PI-ME/CFS (red) or males (turquoise) and females (orange) for the indicated PCs. **c** Venn diagram showing common DE genes identified from male and female cohorts (DE genes are genes with $p$ value < 0.05 using an unadjusted two-sided moderated t-test). DE analysis using an unadjusted two-sided moderated t-test on the (**d**, **e**, **h**, **i**) male cohorts and (**f**, **g**, **j**, **k**) female cohorts. **d**, **f** PCA plot computed from DE genes shows robust clustering of samples based on the PI-ME/CFS status. **e**, **g** Volcano plots shows log transformed statistically significant (unadjusted $p$-values < 0.05 using an unadjusted two-sided moderated t-test) up (red) and down regulated (blue) genes

in PI-ME/CFS male and female cohorts, respectively. Pathway enrichment plot of DE genes (**h**) upregulated and (**i**) downregulated in PI-ME/CFS male cohort. Pathway enrichment plot of DE genes (**j**) upregulated and (**k**) downregulated in PI-ME/CFS female cohorts. The top 15 pathways are labeled on the y axis and the color of the circle is scaled with −log-10 $p$-value and the size of the pathway circles inside the plot are proportional to the number of genes that overlapped with the indicated pathway. Fisher's exact test was used in the *'clusterProfiler'* R package to obtain the log-transformed $p$-values. Red color nodes: upregulated in PI-ME/CFS and blue color nodes: downregulated in PI-ME/CFS. DE differentially expressed, PC principal component. Source data are provided as a Source Data file.

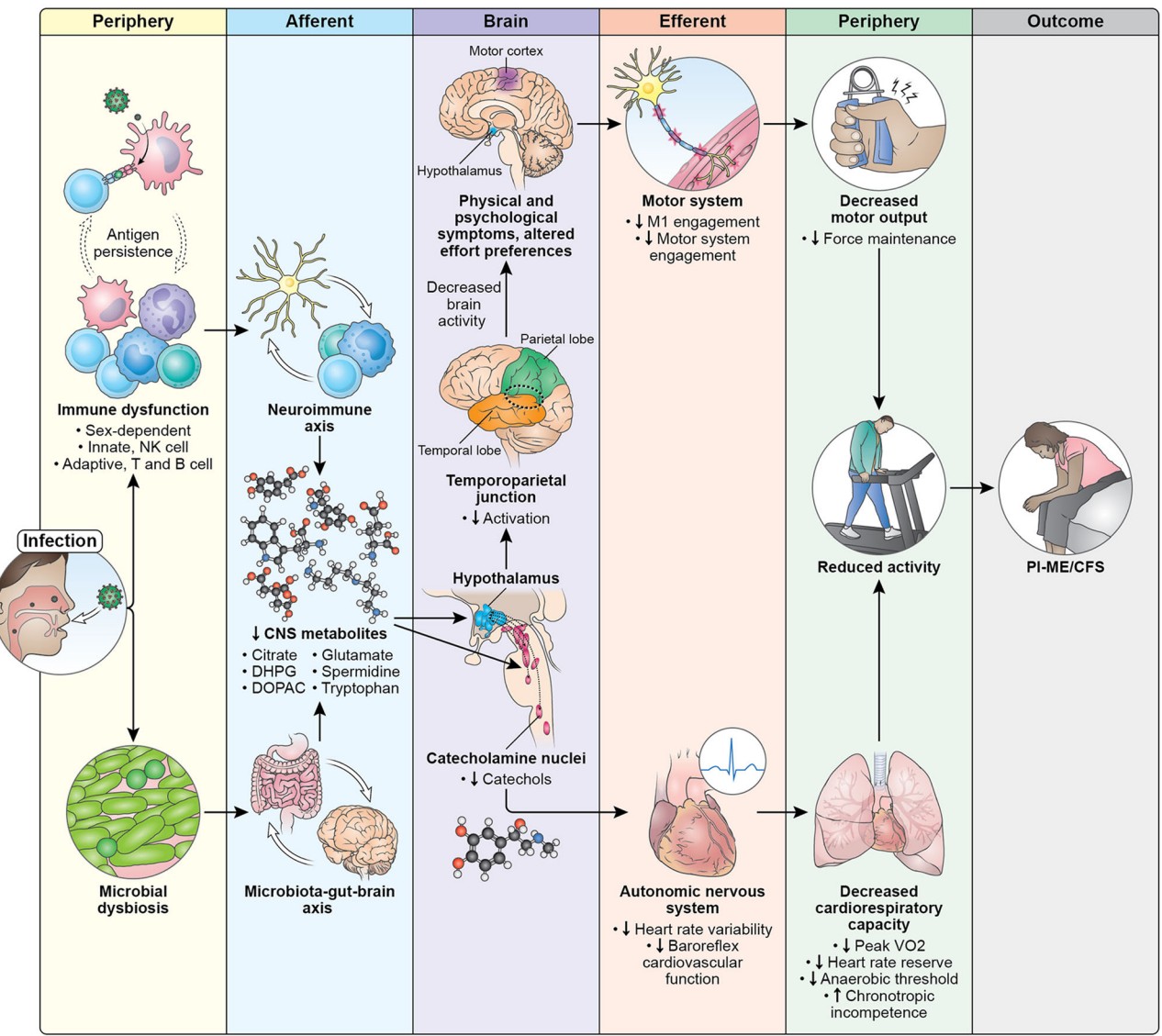

**Fig. 10 | Pathophysiology of PI-ME/CFS.** Diagram illustrates potential mechanisms and a cascade of events that lead to the development of ME/CFS after an infection. Exposure to an infection leads to concomitant and persistent immune dysfunction and changes in gut microbiome. Immune dysfunction affects both innate and adaptive immune systems that are sex dependent. We hypothesize that these changes are driven by antigen persistence of the infectious pathogen. These immune and microbial alterations impact the brain, leading to decreased concentrations of metabolites which impacts brain function. The catecholamine nuclei release lower levels of catechols, which impacts the autonomic nervous system and manifests with decreased heart rate variability and decreased baroreflex

cardiovascular function, with downstream effects on cardiopulmonary capacity. Altered hypothalamic function leads to decreased activation of the temporoparietal junction during motor tasks, suggesting a failure of the integrative brain regions necessary to drive the motor cortex. This decreased brain activity is experienced as physical and psychological symptoms and impacts effort preferences, leading to decreased engagement of the motor system and decreases in maintaining force output during motor tasks. Both the autonomic and central motor dysfunction result in a reduction in physical activity. With time, the reduction in physical activity leads to muscular and cardiovascular deconditioning, and functional disability. All these features make up the PI-ME/CFS phenotype.

decreased brain activity is experienced as physical and psychological symptoms and impacts effort preferences, leading to decreased engagement of the motor system and decreases in maintaining force output during motor tasks. Both the autonomic and central motor dysfunction result in a reduction in physical activity. With time, the reduction in physical activity leads to muscular and cardiovascular deconditioning, and functional disability. These features make up the PI-ME/CFS phenotype.

This model suggests places for potential therapeutic intervention and explains why other therapies have failed. The finding of possible immune exhaustion suggests that immune checkpoint inhibitors may be therapeutic by promoting clearance of foreign antigen. Immune dysfunction leads to neurochemical alterations that impact neuronal circuits, which may be another point of intervention. Therapeutically targeting downstream mechanisms, with exercise, cognitive behavioral therapy, or autonomic directed therapies, may have limited impact on symptom burden, as it would not address the root cause of PI-ME/CFS. However, combination therapy affecting multiple pathways could be considered. The finding of substantial physiological differences related to sex suggest that there may not be a single unified mechanism that leads to PI-ME/CFS and that successful therapy may require a personalized medicine approach.

In conclusion, PI-ME/CFS is a distinct entity characterized by somatic and cognitive complaints that are centrally mediated. Fatigue is defined by effort preferences and central autonomic dysfunction. There are distinct sex signatures of immune and metabolic dysregulation which suggest persistent antigenic stimulation. Physical deconditioning over time is an important consequence. These findings identify potential therapeutic targets for PI-ME/CFS.

## Methods

### Ethics statement
All research procedures were approved by the NIH Central IRB (NCT 02669212) and performed in accordance with the Declaration of Helsinki. Informed consent was obtained from all study participants.

### Recruitment and screening
The PI-ME/CFS group was selected based on medical record documentation of persistent and severe fatigue and post-exertional malaise as the consequence of an acute infection within the last five years without a prior history of explanatory medical or psychiatric illness. The full inclusion/exclusion criteria can be found in Supplementary Data S1.

Potential ME/CFS participants initially completed a telephone screening interview. Those passing the initial screen were contacted for a physician telephone interview and medical record review. This review process was iterative, completing when adequate documentation of infection and ME/CFS was provided, when exclusionary documentation was noted, or when all available medical records were exhausted. Only participants with adequate documentation were invited to NIH for a phenotyping visit for case ascertainment and to collect research measures.

The healthy volunteer (HV) group consisted of demographically matched persons without clinical fatigue and free from medical disease. HVs were recruited from referrals from the NIH Office of Patient Recruitment and responses to study advertisements. HVs were admitted to the NIH Clinical Center for a phenotyping visit to rule out occult illness and to collect research measures. HVs without occult illness were invited to return for an additional exercise stress visit.

Demographic information, including age, birth sex, and gender, for all participants can be found in the Source Data file for Fig. 1b. Birth sex and gender information were collected by self-report during an intake interview by the study physician (BW). ME/CFS is reported to occur three times more frequently in females and efforts were made in recruitment to obtain a balanced birth sex distribution. All participants

were compensated consistent with National Institutes of Health guidelines.

### Case ascertainment
Participants who potentially qualified based on screening were admitted to the NIH Clinical Center for a week-long case ascertainment visit. A detailed history and physical was performed on each participant in duplicate. An internal medicine nurse practitioner and a rheumatologist performed separate evaluations and consulted together after both were completed. A complete standardized neurological evaluation was performed by a board-certified neurologist. A psychiatric evaluation was performed by a licensed psychologist. Medical consultants were engaged to evaluate participants when appropriate. Laboratory and imaging studies were performed to investigate potential health issues noted during these medical evaluations. Clinical laboratory testing of blood and cerebrospinal fluid, brain magnetic resonance imaging, polysomnography, and an orthostatic challenge were also used in case ascertainment and are described in more detail below.

We excluded participants taking systemic immunomodulatory drugs. All participants were on stable medication dosages throughout the study. Medications that would interfere with study measurements were tapered off for a minimum of three half-lives prior to collection.

### Case adjudication
Clinical information from the visit was compiled and reviewed by a Case Adjudication panel. Adjudicators were all recognized clinical experts in ME/CFS (Lucinda Bateman, Andy Kogelnik, Anthony Komaroff, Benjamin Natelson, Daniel Peterson). Each adjudicator performed their own independent review to assign both ME/CFS case status and temporality of ME/CFS onset to an infection. When discrepancies arose between adjudicators, a case adjudication meeting was convened. Adjudicators had to unanimously agree that a participant developed ME/CFS after a documented infection for a case to be considered adjudicated and included in the analyses. Positively adjudicated participants were also invited to return for an additional 10-day long exercise stress visit.

To be considered an adjudicated case, participants were required to be unanimously considered to be a case of PI-ME/CFS by the protocol's adjudication committee, meet at least one of three ME/CFS criteria (1994 Fukuda Criteria[64], 2003 Canadian Consensus Criteria for Myalgic Encephalomyelitis/Chronic Fatigue Syndrome[65], or the Institute of Medicine Diagnostic Criteria[66]), have moderate to severe clinical symptom severity as determined by having a Multidimensional Fatigue Inventory (MFI) score of ≥13 on the general fatigue subscale or ≥10 on the reduced activity subscale, and functional impairment as determined by having a Short-Form 36 (SF-36) score of ≤70 on physical function subscale, or ≤50 on role physical subscale, or ≤75 on social function subscale.

### Life narratives and qualitative interviews
Detailed life narratives and qualitative interviews were conducted to understand the lived experience of PI-ME/CFS, the context of the life it occurred in, and to capture the point-in-time experience of post-exertional malaise. Participants had a life narrative collected by a study investigator, an interview about the impact of PI-ME/CFS by an occupational therapist, and up to 10 brief semi-structured qualitative interviews that were developed based on data collected from a preparatory focus group study[34]. Evocative quotes from the transcripts were selected by the investigators to give a patient voice to match the findings reported.

### Performance validity testing
Symptom validity tests were administered to determine if subjective reporting was consistent within individuals, to identify atypical

patterns of responding, and to relate these measures to population norms. Performance validity tests were administered to determine whether neuropsychological test performances completed during the study were responded to validly, that is, they reflect neurocognitive functioning and are not unduly impacted by non-cognitive factors, such as poor engagement in testing. These tests include: Minnesota Multiphasic Personality Inventory – 2 Restructured Form (MPI2-RF): examinees have to complete true-false items that best describes themselves and was scored using various empirically-derived validity indices;[67] B Test: examinees have to rapidly discriminate between letter stimuli;[68] Dot Counting Test: examinees have to count dots as rapidly as possible;[69] Word Memory Test: a task where examinees have to view words and later remember them[70].

### Patient reported outcomes measures

**The following symptom and health questionnaires were administered.** Short-Form 36 (SF-36): a standard measure of health-related quality of life outcomes that has been tested and validated extensively in a number of clinical populations;[71] CDC Symptom Inventory (CDC-SI): collects information on occurrence, frequency, and intensity of symptoms common in ME/CFS and other fatiguing illnesses;[72] Multidimensional Fatigue Inventory (MFI): a validated 20-item self-report instrument designed to measure fatigue severity;[73] Patient Reported Outcomes Measurement Information System−Short Forms (PROMIS-SF): a system of highly reliable, precise measures of patient−reported health status for physical, mental, and social wellbeing. PROMIS forms administered included Fatigue, Pain Behavior, Pain Interference, Pain Intensity, Global Health, Emotional Distress−Anxiety, Emotional Distress- Depression, Sleep-Related Impairment and Sleep Disturbances;[74] The McGill Pain Questionnaire (MPQ): a list of 20 groups of adjectives to describe sensory, affective and evaluative aspects of pain;[75] The Neuropathic Pain Scale (NPS): a questionnaire designed to assess the quality and the intensity of the neuropathic sensations;[76] Poly-symptomatic Distress Scale: a self-administered instrument that determines both the distribution of painful areas across the body and an estimate of related symptom burden that can be used to define fibromyalgia;[77] Patient Health Questionnaire-15 (PHQ-15): a validated questionnaire used to assess somatic symptom severity and the potential presence of somatization and somatoform disorders;[78] Pittsburgh Sleep Quality Index (PSQI): a measure of sleep quality over a 1-month period;[79] Fatigue Catastrophizing Scale: a measure of catastrophizing related to the fatigue experience;[80] The Multiple Ability Self-Report Questionnaire (MASQ): a questionnaire that assesses the subjective appraisal of cognitive difficulties in five cognitive domains: language, visual-perceptual ability, verbal memory, visual-spatial memory, and attention/concentration;[81] and Belief about Emotions scale: A validated questionnaire designed to measure the beliefs regarding expressing negative thoughts and feelings[82].

### Brain magnetic resonance imaging

MRI was obtained on a 3.0 tesla Philips Achieva device. Sequences performed precontrast were 2D axial proton density-and T2-weighted, 15 direction diffusion tensor imaging (DTI) with b = 1000 DTI. 3D sagittal T1 magnetization prepared rapid acquisition gradient echo (MPRAGE), and T2 weighted fluid attenuated inversion recovery (FLAIR) with approximately 1 mm isotropic resolution. Gadolinium based contrast agent was injected slowly over approximately one minute while high resolution 0.55 isotropic susceptibility weighted imaging was obtained. Following this post contrast images were obtained using 3D sagittal T1 fast field echo (FFE) and T2 Weighted FLAIR techniques. DTI data was processed to generate diffusion weighted imaging (DWI) and apparent diffusion coefficient (ADC) images. Scans were evaluated in duplicate. A neuroradiologist and a neurologist performed separate evaluations. Differences in interpretation were resolved through consultation.

### Small fiber density measures

Two 3-mm excisional skin biopsies were collected from the distal thigh and distal leg. Samples were fixed, sectioned and immunostained for the panaxonal marker PGP9.5 by free-floating immunohistochemistry. Four skin sections from each biopsy were randomly selected, immunostained, and mounted on a single slide and epidermal nerve fibers were visualized with confocal microscopy. This method provides an accurate representation of the biopsy sample while avoiding sampling error[83]. A diagnosis of small fiber sensory neuropathy is given based upon a length-dependent loss of epidermal nerve fibers.

### Measures of brain injury

Plasma and cerebrospinal fluid were analyzed for markers of brain injury by immunoassay using digital array technology, which uses a single molecule enzyme-linked immunoarray (Simoa) method[84]. The Neurology 4-Plex A platform for Nf-L, Tau, GFAP, and UCH-L1 was used. In brief, paramagnetic capture beads coated with each relevant antibody, and a biotinylated detector for each relevant antibody are combined. Antibody coated paramagnetic capture beads and labeled biotinylated detector antibody bind to the relevant molecules present in the sample. Following a washing step, a conjugate of streptavidin-beta-galactosidase (SBG) is mixed with the capture beads. The captured molecules become enzymatically labeled when the SBG binds to the biotinylated detector antibodies. A second wash is performed, and the capture beads are resuspended in a resorufin beta-D-galactopyranoside (RGP) substrate solution. This suspension is transferred to the Simoa Disc. Individual paramagnetic capture beads settle into 216,000 femtoliter-sized microwells designed to hold no more than one bead per well. The beads are sealed into the microwells while excess beads are sealed into the microwells while excess beads are washed away with a synthetic fluorinated polymer sealing oil. If the measured molecule is present in the sample and subsequently captured and labeled, the beta-galactosidase hydrolyzes the RGP substrate and produces a fluorescent signal. This signal is detected and counted by the Simoa optical system. The concentrations of relevant molecules are interpolated from a standard curve.

### Clinical laboratory measurements of blood and cerebrospinal fluid

The following panel of laboratory evaluations were performed on collected blood samples: acute care panel, mineral panel, hepatic panel, complete blood count with differential, prothrombin time, international normalized ratio, partial thromboplastin time, thyroid stimulating hormone, free thyroxine, triiodothyronine, iron, ferritin, transferrin saturation, fasting lipid panel, hemoglobin A1c, anti-nuclear antibody, Rheumatoid factor, anti-cyclic citrullinated antibody, anti-Smith antibody, anti-RNP, ssA, ssB, vitamin B12, 25(OH) vitamin D, $1,25(OH)_2$ vitamin D, folate, creatine kinase, c-reactive protein, erythrocyte sedimentation rate, d-dimer, quantitative immunoglobulins, flow cytometry for lymphocyte subsets, human immunodeficiency virus by enzyme-linked immunosorbent assay, Epstein-Barr virus and Cytomegalovirus by polymerase chain reaction, Epstein-Barr antibodies, C6 peptide antibodies, hepatitis panel, rapid plasma regain, and tryptase level.

Heavy metal screening was performed on urine samples collected over 24-hours in a CLIA certified laboratory using inductively-coupled plasma/mass spectrometry (ICP/MS).

Cerebrospinal fluid samples were analyzed for cell counts, glucose, protein, and oligoclonal bands.

### Cerebrospinal fluid ICP-MS (inductively coupled plasma-mass spectroscopy)

ICP-MS is an analytical technique by which concentrations of elements are determined up to as low as ppt (parts per trillion) levels in liquid, solid or gaseous samples. Elements are led through a plasma source

where the atomic forms of elements become ionized. These ions are then detected according to their masses.

Total iron concentrations in the cerebrospinal fluid samples were measured by ICP-MS (Agilent model 7900). For each sample, 200 μL of concentrated trace-metal-grade nitric acid (Fisher) was added to 200 μL of sample taken in a 15 mL Falcon tube. Tubes were sealed with electrical tape to prevent evaporation, taken inside a 1 L glass beaker, and then placed at 90 °C oven. After overnight digestion, each sample was diluted to a total volume of 4 mL with deionized water, and then analyzed by ICP-MS.

### Dietary evaluation
Participants completed the Diet History Questionnaire (DHQII), an internet-based survey that asks 134 questions regarding dietary intake over the past year and eight questions about dietary supplement intake. Participants kept seven-day food records, which were reviewed by nutrition staff and coded into Nutrition Data Systems for Research (NDSR) software to obtain nutrient intake data.

### Medication reconciliation
Medication and supplement use were collected during the history and physical exam.

### Sleep measurements
PI-ME/CFS participants each had a standard clinical polysomnogram to evaluate for obstructive sleep apnea, periodic limb movements, and sleep fragmentation.

### Heart rate variability
Standard three-lead ambulatory ECG monitors were used for 24-h recordings. All data were downloaded on-site and reviewed by ECG telemetry nurses and by a pediatric cardiologist with clinical electrophysiology training. The recorded data was analyzed using Spacelabs Impresario (version 3.07.0158) program. Arrhythmia and non-normal beats were detected, coded appropriately and excluded from subsequent HRV analysis. Tracings were reviewed for electrical and mechanical noise artifact and these portions of tracings were similarly excluded from subsequent analysis. Recordings with less than 22 h of data were excluded.

### Orthostatic challenge
Participants were fitted with electrodes to measure cardiac signals and electrical impedance, a respiratory belt, a pulse oximeter, a finger-cuff for beat-to-beat blood pressure measurements, an automated blood pressure cuff, and a forearm plethysmograph transducer paired with a rapid-inflation brachial cuff for forearm blood flow measurements.

Prior to the orthostatic challenge, baseline hemodynamic measures and a blood sample were collected. The participant was then tilted head-up at a 70-degree angle. The orthostatic challenge was continued for 40 min, with hemodynamic information collected in real time and blood samples collected at four minute intervals. The orthostatic challenge was ended if a participant developed hemodynamic instability or acute symptoms. On completion of the orthostatic challenge, the participant was returned to a supine position for 10 min, at which time final hemodynamic and blood measures were made.

### Baroreflex function measurements
Participants were fitted with electrodes to measure cardiac signals and electrical impedance, a respiratory belt, a pulse oximeter, a finger-cuff for beat-to-beat blood pressure measurements, an automated blood pressure cuff, and a forearm plethysmograph transducer paired with a rapid-inflation brachial cuff for forearm blood flow measurements. Deep breathing measures were collected with the participant supine breathing deeply at a rate of five to six breaths per minute for three minutes. Three or more Valsalva maneuvers were then performed,

during which the participants blow against resistance for 12 s at 30 mmHg and then relaxes. For participants where a square wave phenomenon was observed, the participant was tilted at 20 degrees head up and the procedure repeated.

### Cerebrospinal catechol measurements
Cerebrospinal levels of catechols were assayed by batch alumina extraction followed by liquid chromatography with series electrochemical detection as reported previously[85,86]. In summary, to freshly thawed CSF in a plastic sample tube, approximately 5 mg of acid washed alumina, internal standard, and TRIS/EDTA buffer were added for alumina adsorption of the catechols. The tube was shaken vigorously using a paint can shaker for about 20 min. The tube was then spun in a microfuge, the alumina forming a pellet at the bottom of the tube. The supernatant was removed, and the alumina was washed twice. Then, after removal of the supernatant, 100 μL of an acidic eluting solution was added to the tube for desorbing the catechols from the alumina. The tube was shaken in a vortex mixer and then centrifuged. The supernatant was removed manually using a pipette and transferred to a microvial and placed in the carousel of the automated injector. For most samples 90 μL was injected onto the liquid chromatography column. The column eluate was passed through 3 electrodes in series, the first set at an oxidizing potential and the third set at a reducing potential. The electrochemical signal from the reducing electrode was recorded using proprietary software, peak heights of compounds with retention times of interest were measured, and concentrations of analytes in units of pg/mL were tabulated in a spreadsheet using a macro after adjustment for analytical recovery of the internal standard. For reporting purposes concentration in pmol/mL were used.

### Psychiatric evaluation
The following psychological inventories were administered: Composite International Diagnostic Interview Trauma Section (CIDI-Trauma): a validated survey that characterizes a participant's previous traumatic experiences;[87] Post-traumatic Stress Diagnostic Scale (PDS): a validated instrument for the epidemiologic diagnosis of Post-traumatic Stress Disorder;[88] Childhood Trauma Questionnaire Short Form (CTQ-SF): a validated instrument that characterizes potential traumatic life experiences in early childhood;[89] Sexual and Physical Abuse Questionnaire (SPAQ): a validated questionnaire that characterizes the type and age of occurrence of traumatic life experiences;[90] Beck Depression Inventory −II (BDI-II): a validated self-report inventory for measuring the severity of depression;[91] Beck Anxiety Inventory (BAI): a validated self-report inventory for measuring the severity of anxiety[92] Center for Epidemiologic Studies Depression Scale−Revised (CESD-R): A validated self-report inventory for screening for depression; Minnesota Multiphasic Personality Inventory – 2 Restructured Form (MMPI2-RF): examinees have to complete true-false items that best describe themselves; Structured Clinical Interview−DSM 5 (SCID-5): History of current and past psychiatric diagnosis was assessed with the Structured Clinical Interview for DSM-5, Research Version (SCID-5-RV).

### Body composition measurements
Weight (Scale-Tronix 5702 digital balance, Carol Stream, IL, USA) and height (Seca 242 stadiometer, Hanover, MD, USA) were taken at fasted conditions. Body composition, including body fat mass, lean soft tissue mass, and fat percentage was measured by dual-energy X-ray absorptiometry (iDXA scanner with Encore 15.0 software; GE Healthcare, Madison, WI, USA).

### Mitochondrial extracellular flux testing
Mitochondrial function was assessed in PBMC that were isolated and measured within three hours of being collected using an extracellular

flux assay (Mito Stress Test, Agilent)[93]. In brief, PBMCs were isolated from 8 ml of blood. Samples were centrifuged at $1,750 \times g$ for 30 min at room temperature. The cloudy layer was transferred to a 15 mL conical tube where 15 mL of PBS was added and inverted five times. The sample was then centrifuged at $300 \times g$ for 15 min at 4 °C. After discarding the supernatant, the pellet was re-suspended by adding 10 mL PBS and inverting five times. The sample was then centrifuged at $300 \times g$ for 10 min at 4 °C. After discarding the supernatant, the cells were re-suspended in 1 mL PBS and centrifuged at $610 \times g$ for 10 min. After removing the supernatant, the pellet was re-suspended in complete RPMI-1640 supplemented with 10% FBS, 10 mM Penicillin/Streptomycin. Cell plates were coated with tissue adhesive solution after diluting the stock solution in 0.1 M sodium bicarbonate pH 8.0 solution. 100 μL of the diluted solution was added to each well and incubated for 20 min at room temperature, washed with deionized water and air dried. On the day of the experiment, fresh assay media was prepared by adding L-glutamine, pyruvate, and glucose to base media (the same constituents as Dulbecco's Modified Eagle's Medium (DMEM), but without any sodium bicarbonate, glucose, glutamine, or sodium pyruvate) to make assay media and warmed to 37 °C and adjusted to a pH of 7.4. PBMCs were plated into each well to reach 80–90% confluency. Plates were centrifuged at $200 \times g$ for 2 min, then washed with assay media. 180 μL of assay media was then added to each well and incubated in a non-CO2 37 °C incubator for 60 min prior to performing the extraceullar flux assay according to manufacturer's directions. Assay Results of the assay were normalized to amount of live cells in the cell preparation[93].

## ATP 9.4 characterization of muscle

Samples were collected from the vastus lateralis muscle and flash frozen. Frozen sections of muscle samples were prepared with cryostat, 10um thickness. Six slides of frozen sections were taken to Johns Hopkins University, Neurology department, Neuromuscular Laboratory for ATPase pH 9.4 stain, which can identify Type II muscle fibers.

The calcium method for myosin-ATPase demonstration, employing solutions of different pH values, has been used primarily to distinguish muscle fiber types. Muscle fibers may be broadly categorized as type I (slow, red muscle, oxidative) and type II (fast, white muscle, glycolytic). Type II muscle fibers are further subdivided as IIa (glycolytic), IIb (glycolytic/oxidative), and IIc which may be fibers that are changing types due to disease, injury, or development. The pre-incubation pH relatively inactivates the myosin-ATPase iso-enzyme characteristic of specific fiber types. The remaining active ATP and calcium-dependent enzyme activity releases calcium atoms which are replaced by cobalt, and finally precipitated as a black insoluble cobalt salt of ammonium sulfide. Slides were scanned at National Institute of Arthritis and Musculoskeletal and Skin Diseases (NIAMS) Light Image department and images, 2.5X size, were acquired using NDP.view2 software from Hamamatsu.

Analysis of images was performed using Fiji (Image J) in a semi-automated way to facilitate the evaluation of cross-sectioned myofibers in the largest possible area for each image. Preprocessing and thresholding of the images using Fiji available filters and tools was first done to generate a selection of muscle fibers and to classify them according to the intensity of the staining; however, the accuracy of this evaluation was limited, and human intervention was required during the analysis. Following automatic fiber selection, a researcher blinded to sample identity modified the selection of fibers to exclude regions of the image where fibers were longitudinal or where the sample quality was suboptimal. In addition, fiber segmentation was manually corrected to properly delineate a region of interest for each fiber where automatic delimitation was not accurate. Similarly, automatic classification of the fibers was manually reviewed. The minimum Feret diameter was measured for each fiber, and the median of the measurements for type I and type II fibers in each image was calculated and

used to compute a Type II/Type I ratio to assess the relative fiber size for the image.

## Mitochondrial genetic analysis

Samples were collected from the vastus lateralis muscle and flash frozen. Sample analysis was performed by GeneDx.

Genomic DNA was extracted from the specimens. For the nuclear genome, the DNA is enriched for the complete coding regions and splice junctions of the genes on this panel using a proprietary targeted capture system developed by GeneDx for next-generation sequencing with CNV calling (NGS-CNV). The enriched targets were simultaneously sequenced with paired end reads on an Illumina platform. Bi-directional sequence reads were then assembled and aligned to reference sequences based on NCBI RefSeq transcripts and human genome build GRCh37/UCSC hg19. After gene specific filtering, data were analyzed to identify sequence variants and most deletions and duplications involving coding exons; however, technical limitations and inherent sequence properties effectively reduce this resolution for some genes. Alternative sequencing or copy number detection methods were used to analyze regions with inadequate sequence or copy number data by NGS. The entire mitochondrial genome from the submitted sample was also amplified and sequenced using Next Generation sequencing. DNA sequence was assembled and analyzed in comparison with the revised Cambridge Reference Sequence (rCRS GeneBank number NC_012920) and the reported variants listed in the MITOMAP database (http://www.mitomap.org). Next generation sequencing may not detect large-scale mtDNA deletions present at 5% heteroplasmy or lower or mtDNA point variants present at 1.5% heteroplasmy or lower. Reportable variants include pathogenic variants, likely pathogenic variants and variants of uncertain significance. Likely benign and benign variants, if present, were not reported. For the nuclear genome, the technical sensitivity of sequencing is estimated to be >99% at detecting single nucleotide events. It will not reliably detect deletions greater than 20 base pairs, insertions or rearrangements greater than 10 base pairs, or low-level mosaicism. The copy number assessment methods used with this test cannot reliably detect copy number variants of less than 500 base pairs or mosaicism and cannot identify balanced chromosome aberrations. Assessment of exon-level copy number events is dependent on the inherent sequence properties of the targeted regions, including shared homology and exon size. Due to the presence of non-functional pseudogenes, regions of the GYG2, NR2F1, PDSS1, and TSFM, and genes are not fully sequenced by this method. For the COQ7, COX8A, HTRA2, NDUFB11, RNASEH1, SCO2, SDHA, SLC25A26, SLC25A46, TFAM, TMEM126B, and TRMT10C genes, sequencing but not deletion/duplication analysis was performed. In addition, the COA5 gene deletion/duplication analysis may only be able to detect full gene events. For the mitochondrial genome, next generation sequencing can detect mtDNA point variants as low as 1.5% heteroplasmy and large-scale deletions (2 kb or larger) as low as 5% heteroplasmy. However, for large-scale mtDNA deletions observed at less than 15% heteroplasmy, a quantitative value will not be provided. This test is expected to detect greater than 98% of known pathogenic variants and deletions of the mitochondrial genome.

The mitochondrial variants identified by GeneDx were annotated using Annovar[94].

## Modified effort expenditure for rewards task

The Effort-Expenditure for Rewards Task[15] is a multi-trial game in which participants were offered a choice between two task difficulty levels for a reward (Supplementary Fig. S5A). The task began with a one second blank screen, followed by a five second choice period in which the participant was informed of the value of the reward and the probability of winning if the task were completed successfully. After the participant chose the task, another one second blank screen was displayed. Next, the participant either completed 30 button presses in

seven seconds with the dominant index finger if they chose the easy task, or 98 button presses in 21 s using the non-dominant little finger if they chose the hard task. Next, the participant received feedback on whether they completed the task successfully. Finally, the participant learned if they have won, based upon the probability of winning and the successful completion of the task. This process repeats in its entirety for 15 min.

Participants were told at the beginning of the game that they would win the dollar amount they received from two of their winning tasks, chosen at random by the computer program (range of total winnings is $2.00–$8.42).

The primary measure of the EEfRT task is Proportion of Hard Task Choices (effort preference). This behavioral measure is the ratio of the number of times the hard task was selected compared to the number of times the easy task was selected. This metric is used to estimate effort preference, the decision to avoid the harder task when decision-making is unsupervised and reward values and probabilities of receiving a reward are standardized.

## Actigraphy
Participants were instructed to wear ActiGraph GT3X+ accelerometers (ActiGraph Inc., Pensacola, FL, USA) on their waist and non-dominant wrist, continuously, for at least seven consecutive days at home. Raw tri-axial accelerometer data was recorded at 80 samples/second and subsequently filtered and aggregated into one-minute vector magnitude activity counts and steps with Actilife software (v6.13.0, Acti-Graph, Pensacola, FL, USA) and customized programs in Matlab (R2021b, Mathworks, Natick, MA, USA). Periods of sixty or more consecutive minutes of zero vector magnitude counts were identified as non-wear, and daily data were considered valid if the device was worn for ≥10 h from 12 midnight to 12 midnight the next day[95]. Minutes where activity counts of the vertical axis of the waist-worn accelerometer fell between 2020 and 5998 were identified as moderate intensity activity, i.e., three to six times the resting metabolism.

## Grip strength
Grip strength was measured with a hand-held dynamometer (Jamar). Each hand was tested individually with the arm, forearm, and wrist in a neutral position. First, each participant was instructed to exert a maximum possible grip force for about five seconds. After completing the first reading, this maximal grip task was repeated. After a minute of rest, the participant was asked to maintain their maximal grip for as long as possible. The time elapsed when the participant's grip force reduced to 50% of their maximum was recorded by an investigator. All three tests were then repeated with the other hand.

## Electrophysiology and repetitive grip testing
To assess physical fatigue, a grip force task was designed which required participants to try to maintain their grip at 50% of maximum voluntary contraction (MVC) in successive blocks of 30 s interspaced with 30 s rest blocks. Each participant sat with their right forearm placed in a rigid-frame dynamometer (biopac, Goleta, CA, USA). The MVC was set as the highest value of three squeezes.

Electromyography (EMG) was collected using surface electrodes (3 M, St. Paul, MN, USA) over the flexor and extensor carpi radialis (FCR, ECR) muscles and the abductor pollicis brevis (APB), using amplifiers and software from Cambridge Electronic Design, Cambridge, UK.

Transcranial magnetic stimulation (TMS) was performed to probe the excitability of the primary motor cortex (m1) via motor evoked potentials (MEPs). A 70 mm figure-8 coil was used to determine the optimal position for evoking MEPs by holding the coil tangential to the scalp and slightly displacing it until the highest MEP amplitude was recorded in the APB muscle. The positions of the participant's head and TMS coil were tracked with a neuronavigation system (rogue

Research, Cambridge, MA, USA) in order to maintain the stimulation position over the hotspot. The TMS Input-Output curve was recorded, collecting MEP responses for intensities 5–100%, in increments of 5%, of stimulation output, in order to calculate the S50. The S50 value was defined as 50% of maximum MEP amplitude. This curve value was also used to estimate resting motor threshold (rMT), which was confirmed using single pulse TMS as the stimulation intensity that would evoke a ~50 μV response in roughly 50% of pulses delivered.

The maximum M-wave was also determined prior and after the repetitive grip task by applying electrical stimulation over the nerve innervating the FCR muscle.

During the repetitive grip task, each participant repeatedly performed 30-s periods of isometric muscle contractions aiming at 50% of MVC. Generally, participants performed 16 blocks, but some quit earlier, and some continued for up to 32 blocks. After each squeeze block, there was a 30 s period of rest. During this rest period, MEPs (elicited every five seconds) were measured.

The development of muscular fatigue during the task, defined as the inability to maintain at least 40% MVC force for more than three seconds, was analyzed by comparing the 1st block (no fatigue), the last block before fatigue onset, and three following blocks after fatigue onset or, if they did not fatigue, the last four blocks. For EMG, we used the Dimitrov index (DI)[17,18] to evaluate the shift in EMG frequency power within blocks.

## Functional MRI repetitive grip testing
Brain activity was also assessed during the grip strength task with fMRI. This task was designed to identify, at the whole brain level, brain areas involved in fatigue. Participants lay supine in the scanner and performed repeated 30-s blocks of grip strength with a dynamometer (isometric muscle contractions) at 50% of their maximum voluntary contraction (MVC) interspaced with 30-s blocks of rest. MVC of the forearm muscles was determined from the best of three brief squeezes on the dynamometer. Participants used visual feedback from a computer to monitor force generation. Similar to the TMS study, participants performed 16 blocks, but some quit earlier, and some continued for up to 32 blocks. Subjective appraisals of muscular fatigue were measured with a VAS before and after the grip strength task.

We used a 3 T Prisma SIEMENS scanner equipped with a 64-channel head-coil in the Nuclear Magnetic Resonance Center at the National Institutes of Health. We acquired T2*-weighted EPI with TR = 2 s, TE = 30 ms, image matrix = 64 × 64, flip angle: 70˚, FoV: 100, voxel size 3.5 ×3.5 ×3.5 mm.

## Cardiopulmonary exercise testing (CPET)
CPET was performed using a cycle ergometer and a computerized metabolic cart (CardiO2 Ultima; MedGraphics Corp, St.Paul, MN, USA). A ramp protocol was used where the work rate would be gradually increased until volitional fatigue was reached by each participant. A time-matching paradigm to ensure all participants exercised for between eight to twelve minutes was employed, as per American College of Sports Medicine recommendations[96]. The target endpoint was exertional intolerance defined as the participants' expressed desire to stop cycling despite strong verbal encouragement from the testing staff. Endpoints for stopping the tests were those recommended by the American College of Sports Medicine[96].

Breath-by-breath gas exchange and heart rate (by 12-lead ECG) were measured throughout the CPET. Peak oxygen consumption (peak VO2) was calculated as the 20 s average at the end of the CPET. The anaerobic threshold (AT) identified by the metabolic cart was verified by gas exchange analysis methods[22]. Chronotropic incompetence (CI) was calculated as % predicted heart rate reserve = [peak HR−resting HR]/[APMHR−resting HR] × 100. The slope of the heart rate response during the CPET was examined by linear fit between 15 to 100% of the CPET time. Expected heart rate responses were also generated using

linear fits between predicted resting and peak heart rate values, matching sex and age of both HV and PI-ME/CFS participants. Muscle oxygenation measurements were also made at the vastus lateralis during the CPET by near infrared spectroscopy (NIRO-200NX, Hamamatsu Photonics, Japan). For further determination of maximal oxygenation values of near infrared spectroscopy measurements, a thigh occlusion test was performed prior to the CPET. Following seated rest, an occlusion cuff (Hokanson Rapid Cuff Inflator; Hokanson Inc., Belleview, WA, USA) was rapidly inflated to and held at 80 mmHg above systolic blood pressure for eight minutes.

## Bioenergetic measurements

The metabolic chamber is a whole-room indirect calorimeter that allows detailed assessment of energy and nutrient balance. Measurements are conducted at stable interior (room) temperature, humidity, and barometric pressure, which are continuously measured. Airtight sampling ports and a four-way air-locking food and specimen passage are designed to allow blood draws and specimen retrievals with minimal disturbance to the chamber environment. Outside air is continuously drawn into the chamber, and the flow rate of air at the outlet is measured using a pneumotachograph with a differential manometer. A fraction of the extracted air is analyzed at one minute intervals for $O_2$ and $CO_2$ concentrations with a thermomagnetic $O_2$ analyzer. This allows for a continuous assessment of oxygen consumption ($V\dot{}O_2$), carbon dioxide elimination ($V\dot{}CO_2$), and calculation of overall energy expenditure (EE). The ratio between $V\dot{}CO_2$ and $V\dot{}O_2$ (the respiratory quotient [RQ]) reflects preference for carbohydrate or fat oxidation.

Starting the day prior to CPET, each participant was placed on a metabolic diet controlled for energy and macronutrient content. Measures of energy expenditure and respiratory exchange were obtained during a 16 h (4 pm to 8 am) stay in a metabolic chamber prior to CPET and for three consecutive days afterwards.

## Salivary cortisol measurements

Saliva was collected using a SARSTEDT salivette. Participants did not eat for at least 2 h or drink water 30 min prior to collection. Samples were centrifuged for 2 min at $1000 \times g$ and then frozen at $-80 °C$. Samples were measured by Salimetrics using enzyme-linked immunoassays for cortisol performed in duplicate. Assay sensitivity is <0.007 ug/dL.

## Neuropsychological Measures

The following neuropsychological measures were administered by a trained neuropsychometrist in the following general order: Visual Analogue Scale (Time 1): a set of scales that were administered to capture subjective effort, performance, mental fatigue, and physical fatigue. This test was immediately prior to administration of the neuropsychological testing battery; Wechsler Test of Adult Reading (WTAR) (The Psychological Corporation, 2001): a task that requires the examinee to read words aloud;[97] Hopkins Verbal Learning Test-Revised Learn (HVLT-R Learn): a task where examinees have to learn a list of words;[98] Grooved Pegboard Test: a task where examinees have to rapidly insert pegs in holes;[99] Wechsler Adult Intelligence Scale–Fourth Edition (WAIS-IV) subtests including Coding, Symbol Search and Digit Span: a task where examinees memorize strings of numbers or complete speeded tasks involving unfamiliar symbols;[100] B Test: examinees have to rapidly discriminate between letter stimuli;[68] Hopkins Verbal Learning Test-Revised Delayed Recall (HVLT-R DR): a task where examinees have to recall the list of words previously learned;[98] Brief Visual Memory Test-Revised (BVMT-R): a task where examinees have to learn a list of designs;[101] Visual Analogue Scale (Time 2): scales to capture subjective effort, performance, mental fatigue, and physical fatigue were collected at this time, approximately one hour after testing started; Wisconsin Card Sort Test (WCST-64): a task where examinees have to utilize corrective feedback to learn how to sort cards;[102] Controlled Oral Word Association Test (COWAT; FAS and Animals): a task where examinees have to generate words to various cues;[103] Paced Auditory Serial Addition Test (PASAT): a task where examinees have to rapidly perform serial addition;[104] Brief Visual Memory Test-Revised (BVMT-R) Delayed Recall: a task where examinees have to recall the prior list of designs;[101] Word Memory Test: a task where examinees have to view words;[70] Test of Variables of Attention: a task where examinees rapidly respond using a button press to certain target stimuli and not distractor stimuli;[105] Visual Analogue Scale (Time 3): scales to capture subjective effort, performance, mental fatigue, and physical fatigue were collected at this time, approximately two hours after testing started; Word Memory Test Delayed Recall: a task where examinees have to recall the words viewed earlier;[70] Dot Counting Test: examinees have to count dots as rapidly as possible;[69] MMPI-−2 RF: As described above; EEfRT test: As described above; Visual Analogue Scale (Time 4): scales to capture subjective effort, performance, mental fatigue, and physical fatigue were collected at this time, approximately three hours after testing started.

## PBMC RNA sequencing

RNA was extracted from PBMC's of participants using miRNeasy Micro Kit (QIAGEN). RNA was quantified using Qubit 3.0 fluorometer (Thermo Fisher Scientific) and its integrity confirmed using an Agilent 2100 Bioanalyzer (Agilent Technologies). Dual index libraries were constructed with at least one unique index per each patient library using the TruSeq Stranded Total RNA HT Kit (Illumina) to enable subsequent pooling of equal quantities of individual libraries. The integrity and ratio of pooled libraries was validated using Miseq system (Illumina); then, paired-end sequencing ($2 \times 75$ base pairs (bp)) was performed on an HiSeq 3000 sequencer (Illumina) with the Illumina HiSeq 3000 SBS Hit.

## Proteomics

Peripheral blood serum was isolated using SST tubes and cryopreserved with corresponding cerebrospinal fluid samples. Proteomic analysis used the SOMAscan 1.3k Assay (SomaLogic). This is an aptamer-based assay able to detect 1305 protein analytes, optimized for analysis of human serum[106,107]. Briefly, aptamers are short single-stranded DNA sequences modified to confer specific binding to target proteins and can be highly multiplexed for discovery of biomarker signatures. The proteins quantified include cytokines, hormones, growth factors, receptors, kinases, proteases, protease inhibitors, and structural proteins. The assay was performed according to manufacturer specifications for each of the serum and cerebrospinal fluid sample types. Briefly, serum samples were assayed at three dilutions (40%, 1%, and 0.005%) with each sample dilution added to a corresponding subset of the 1305 SOMAmer detection reagents binned according to manufacturer's predicted target abundance in serum. Cerebrospinal fluid was run at a single 15% concentration dilution with protease inhibitors and polyanionic competitor reagent added. Data was then inspected using a web tool and subjected to quality control procedures as previously described[108,109].

## Flow cytometry of blood and cerebrospinal fluid

EDTA-treated whole blood and cerebrospinal fluid cells were used for flow cytometric analysis. Cerebrospinal fluid samples were obtained by lumbar puncture and the cerebrospinal cells were collected within an hour by centrifugation. Whole blood or cerebrospinal fluid cells were stained with CD3, CD4, CD8, CD14, CD16, CD19, CD25, CD27, CD45, CD45RA, CD56, CD152, CXCR5, IgD, HLA-DR (all from BD Biosciences), PD-1 (BioLegend) and FoxP3 (eBiosciences), as previously described[110]. All flow cytometric analysis was performed using an LSR II (BD Biosciences). The data were analyzed using FlowJo 10.6 software (FlowJo LLC).

An additional panel of T-cell markers including TIGIT, CD244, and CD226 were performed on a subset of participants. Some of these samples were collected from participants during a second lumbar puncture during a return visit months after the initial sample collected for the analysis above.

All antibodies, clones, catalog numbers, manufacturers, and dilutions used in this study are as follows: anti-human CD3 (clone: UCHT1, Cat. 558117, BD Biosciences, 1:30 dilution (1:100 dilution for CSF cells)) anti-human CD3 (clone: UCHT1, Cat. 555335, BD Biosciences, 1:100 dilution); anti-human CD4 (clone: RPA-T4, Cat. 557922, BD Biosciences, 1:30 dilution (1:100 dilution for CSF cells)); anti-human CD4 (clone: RPA-T4, Cat. 555346, BD Biosciences, 1:100 dilution); anti-human CD8 (clone: SK1, Cat. 341051, BD Biosciences, 1:30 dilution (1:100 dilution for CSF cells)); anti-human CD8 (clone: RPA-T8, Cat. 557746, BD Biosciences, 1:100 dilution) anti-human CD14 (clone: M5E2, Cat. 555399, BD Biosciences, 1:30 dilution (1:100 dilution for CSF cells)); anti-human CD16 (clone: 3G8, Cat. 338426, BD Biosciences, 1:30 dilution (1:100 dilution for CSF cells)); anti-human CD19 (clone: HIB19, Cat. 555413, BD Biosciences, 1:30 dilution (1:100 dilution for CSF cells)); anti-human CD25 (clone: M-A251, Cat. 557741, BD Biosciences, 1:30 dilution (1:100 dilution for CSF cells)); anti-human CD27 (clone: M-T271, Cat. 560222, BD Biosciences, 1:30 dilution (1:100 dilution for CSF cells)); anti-human CD45 (clone: HI30, Cat. 560777, BD Biosciences, 1:30 dilution (1:100 dilution for CSF cells)); anti-human CD45RA (clone: HI100, Cat. 555488, BD Biosciences, 1:30 dilution (1:100 dilution for CSF cells)) anti-human CD56 (clone: B159, Cat. 557747, BD Biosciences, 1:30 dilution (1:100 dilution for CSF cells)); anti-human CD226 (clone: 11A8, Cat. 338318, BioLegend, 1:30 dilution (1:100 dilution for CSF cells)); anti-human CD244 (clone: C1.7, Cat. 329522, BioLegend, 1:30 dilution (1:100 dilution for CSF cells)); anti-human PD-1 (clone: EH12.2H7, Cat. 329906, BioLegend, 1:30 dilution (1:100 dilution for CSF cells)); anti-human CXCR5 (clone: RF8B2, Cat. 558113, BD Biosciences, 1:30 dilution (1:100 dilution for CSF cells)); anti-human HLA-DR (clone: G46-6, Cat. 555811, BD Biosciences, 1:30 dilution (1:100 dilution for CSF cells)); anti-human IgD (clone: IA6-2, Cat. 561302, BD Biosciences, 1:30 dilution (1:100 dilution for CSF cells)); anti-human TIGIT (clone: A15153G, Cat. 372714, BioLegend, 1:30 dilution (1:100 dilution for CSF cells)); anti-human CD127 (clone: HIL-7R-M21, Cat. 560551, BD Biosciences, 1:30 dilution); anti-FOXP3 (clone: 236A/E7, Cat. 17-4777-42, Thermo Fisher Scientific, 1:50 dilution); anti-human CD152 (clone: BNI3, Cat. 555853, BD Biosciences, 1:50 dilution); anti-Stat5 (pY694) (clone: 47/Stat5(pY694), Cat. 612567, BD Biosciences, 1:50 dilution).

Dilutions were determined according to our own staining protocols. All the antibodies used for flow cytometry in this study are commercially available, and their specificities have been well validated by the manufacturers and other users.

## NK cell function measurement

NK cell function was measured in blood within 24 h of collection by a [51]Chromium release assay[111] by the clinical laboratory at Cincinnati Children's Hospital.

## Growth differentiation factor-15 measurement

The GDF15 ELISA was performed using R&D Systems, Minneapolis, MN. Catalog No. DGD150 kit as per manufacturer instructions. The intra-assay variation was 1.8−2.8% and the interassay variation was 4.7−5.6%.

## Luciferase immunoprecipitation assay

The luciferase immunoprecipitation systems (LIPS) assay provides an informative tool to explore serology as evidence of autoimmunity and infectious disease exposure due to its ability to efficiently detect antigenicity antibodies against both conformational and linear epitopes. Here, LIPS was used to assess for the presence of autoantibodies against a small diverse panel of known and potential antigens in the PI-

ME/CFS and healthy volunteer participants. The previously described testing format was used to examine antibodies against the various target molecules included known autoimmune-associated proteins (Ro52, Jo-1, TPO, gastric ATPase, tyrosine hydroxylase), neurological autoantigens (GAD65, LGI1, NMDAR1, MUSK), cytokines (Interferon alpha1, Interleukin-6, CXCR4, TGFB1), muscle proteins (MPZ, PMP22), as well as against several infectious agents (HDV, HEV, Zika virus). Light units were measured in a Berthold LB 960 Centro luminometer (Berthold Technologies, Germany) using coelenterazine or furimazine substrate mix (Promega, Madison, WI). In some cases, control sera samples from known positive control autoimmune patients were used as positive controls.

## Muscle RNA sequencing

RNA was extracted from muscle samples using the TRIzol protocol[112]. The integrity of the RNA was verified using a standard quality metric denominated RNA integrity number (RIN) value using the Agilent 4200 TapeStation system and the concentration was measured using the DeNovix DS-11 spectrophotometer. Five hundred nanograms of RNA were used to prepare the RNA sequencing libraries using the NEBNext Ultra II Directional RNA Library Prep Kit and sequenced using the Illumina NovaSeq 6000 sequencer. Reads were demultiplexed using bcl2fastq v. 2.20.0.

## Metabolomics of cerebrospinal fluid

Metabolomics on cerebrospinal fluid samples was performed using Metabolon's Ultrahigh Performance Liquid Chromatography-Tandem Mass Spectroscopy (UPLC-MS/MS). Samples were prepared using the automated MicroLab STAR® system from Hamilton Company. Several recovery standards were added prior to the first step in the extraction process for QC purposes. To remove protein, dissociate small molecules were bound to protein or trapped in the precipitated protein matrix, and to recover chemically diverse metabolites, proteins were precipitated with methanol under vigorous shaking for two minutes (Glen Mills GenoGrinder 2000) followed by centrifugation. The resulting extract was divided into five fractions: two for analysis by two separate reverse phase (RP)/UPLC-MS/MS methods with positive ion mode electrospray ionization (ESI), one for analysis by RP/UPLC-MS/MS with negative ion mode ESI, one for analysis by HILIC/UPLC-MS/MS with negative ion mode ESI, and one sample was reserved for backup. Samples were placed briefly on a TurboVap® (Zymark) to remove the organic solvent. The sample extracts were stored overnight under nitrogen before preparation for analysis.

All methods utilized a Waters ACQUITY ultra-performance liquid chromatography (UPLC) and a Thermo Scientific Q-Exactive high resolution/accurate mass spectrometer interfaced with a heated electrospray ionization (HESI-II) source and Orbitrap mass analyzer operated at 35,000 mass resolution. The sample extract was dried then reconstituted in solvents compatible to each of the four methods. Each reconstitution solvent contained a series of standards at fixed concentrations to ensure injection and chromatographic consistency. One aliquot was analyzed using acidic positive ion conditions, chromatographically optimized for more hydrophilic compounds. In this method, the extract was gradient eluted from a C18 column (Waters UPLC BEH C18-2.1×100 mm, 1.7 μm) using water and methanol, containing 0.05% perfluoropentanoic acid (PFPA) and 0.1% formic acid (FA). Another aliquot was also analyzed using acidic positive ion conditions; however, it was chromatographically optimized for more hydrophobic compounds. In this method, the extract was gradient eluted from the same afore-mentioned C18 column using methanol, acetonitrile, water, 0.05% PFPA and 0.01% FA and was operated at an overall higher organic content. Another aliquot was analyzed using basic negative ion optimized conditions using a separate dedicated C18 column. The basic extracts were gradient eluted from the column using methanol and water, however, with 6.5 mM Ammonium

Bicarbonate at pH 8. The fourth aliquot was analyzed via negative ionization following elution from a HILIC column (Waters UPLC BEH Amide 2.1 × 150 mm, 1.7 μm) using a gradient consisting of water and acetonitrile with 10 mM Ammonium Formate, pH 10.8. The MS analysis alternated between MS and data-dependent $MS^n$ scans using dynamic exclusion. The scan range varied slightly between methods but covered 70–1000 m/z.

## Lipidomics of plasma

The extraction of lipids from plasma was accomplished following manufacturer's protocol, with slight modifications[113]. To 25 μl of plasma sample, 975 μl water, 2 ml methanol, 900 μl dichloromethane (DCM) and 25 μl internal standard was added. The internal standard was prepared from a kit (Avanti Lipids) developed for the Lipidyzer platform (SCIEX)[114,115]. The samples were then vortexed and allowed to sit at room temperature for 30 min. 1 ml water and 900 μl DCM was then added and samples were vortexed and centrifuged for 10 min at $2000 \times g$. The lower organic layer was removed and placed in a separate collection tube. To the remaining aqueous layer, 1.8 ml DCM is added and the samples were vortexed and centrifuged. The lower organic layer was again removed, added to the previous organic layer, and stream dried under nitrogen. The lipids were reconstituted in 250 μl running solvent (50/50 methanol/DCM 10 mM ammonium acetate) and placed in an autosampler vial for analysis. Quality control (QC) samples and 3 QCs spiked with unlabeled internal standards from the QC Spike Standards kit (SCIEX) were run at the beginning, middle, and end of the samples. The quantitation of the lipid species was done with Lipidomics Workflow Manager (SCIEX) 1.0.5.0, which uses validated DMS-MRM acquisition methods, known internal standard concentrations, and integrated peak areas from the samples to determine the quantity of each lipid species. Lipid species were included in the data analyses if above the limit of quantification in >50% of the participants.

## Microbiome stool sampling procedure

Whole fecal samples were collected in a sterile bowl at the NIH Clinical Center, aliquoted, and immediately frozen at −80 °C. Aliquots were sent to the sequencing laboratory for metagenomic and metabolomic analyses.

## Stool microbiome metagenomics

For metagenomic shotgun sequencing, paired-end libraries were prepared from metagenomic DNA using the Illumina Nextera Flex kit, and then sequenced on the Illumina NovaSeq platform with a 2 ×150 bp length configuration. Bioinformatic analysis of the shotgun metagenomic samples was done using the JAMS_BW package, version 1.7.2, which is available on GitHub at https://github.com/johnmcculloch/JAMS_BW. All code for every step in the bioinformatic analysis from reads to plots is publicly available in this package.

Fastqs from each sample were processed with JAMSalpha[116], which, briefly, entails quality trimming and adapter clipping of raw reads with Trimmomatic 0.36[117]. The reads were then aligned against the human genome with Bowtie2 v2.3.2[118] and unaligned (non-host) reads were then assembled using MEGAHIT v1.2.9[119]. For all samples evaluated, the mean sequencing depth (already discounting host reads) was 6.04 Gbp ± 0.79 Gbp, yielding a mean assembly rate of 94.06% ± 1.93%. Assembly contigs smaller than 500 bp were discarded and taxonomic classification of remaining contigs was obtained through Kraken2[120], with a custom 96-Gb Kraken2 database built using draft and complete genomes of all bacteria, archaea, fungi, viruses, and protozoa available in the NCBI GenBank in January 2022, in addition to human and mouse genomes, built using the JAMSbuildk2db tool of the JAMS package. This JAMS-compatible kraken2 database is available for download through the URL https://hpc.nih.gov/~mccullochja/JAMSdb202201.tar.gz. Functional annotation of contigs was obtained

using Prokka v1.14.6[121]. The sequencing depth of each contig was obtained by aligning reads used for assembly back to the contigs. Taxonomy was expressed as the last known taxon (LKT), being the taxonomically lowest unambiguous classification determined for each query sequence, using Kraken's confidence scoring threshold of 5e−06 (using the --confidence parameter). The relative abundance, expressed in parts-per-million (PPM) for each LKT within each sample was obtained by dividing the number of bp covering all contigs and unassembled reads pertaining to that LKT by the total number of non-host base pairs sequenced for that sample.

## Stool nuclear magnetic resonance spectroscopy

Fecal extractions were based on published recommendations[122]. Samples were thawed on ice and buffer ($^2$H-phosphate buffered saline + 0.01% sodium azide; pH 7.5) was added at 5 ml/1 g of material. Samples were sonicated 2× 30 s (50% power) on ice, vortexed for 1 min, and centrifuged at 8000 rcf for 20 min at 4 °C. Supernatant was added to a 3 kDa filter concentrator to further remove particulates and proteins. The flow through was collected after about an hour which amounted to approximately 600 ul. Trimethylsilylpropanesulfonate was added at 200 uM concentration as a chemical shift standard and concentration reference. NMR spectra were acquired on an 800 MHz Agilent DD2 console equipped with a cryogenically cooled probe. A 1-dimensional NOESY sequence with 4 s acquisition time, 1 s recycle delay, and 100 ms mixing time was used. Chenomx software (Alberta, Canada) was used for processing and spectral analysis. Metaboanalyst was used for statistical comparisons.

## Clinical follow-up

Each participant was recontacted by a physician on the investigative team between 11/2021 and 7/2022 to inquire about changes in their clinical condition.

## Statistical analysis

This research protocol utilized an exploratory design coupled with a broad and deep phenotyping approach. The assembled data represents a multidimensional description of post-infectious ME/CFS cases collected with the intent to generate new hypotheses.

Using strict case criteria and adjudication process we minimized medical misattribution and studied a homogeneous population. Since the deep phenotyping resulted in a large number of measurements, our statistical approach used a modified concept of consilience, the principle that evidence from independent, unrelated sources can converge on strong conclusions. Here, measures were purposely selected to interrogate different facets of immunologic, bioenergetic, and homeostatic physiology and determine if similar results would emerge from the different techniques employed. It also used repetition of measures to aid in the interpretation of data variance and reliability.

This exploratory approach embraces the explanatory power of negative findings. Small sample sizes can be adequate for applying logic to demonstrate that a phenomenon is not related to a particular physiological process[123]. These data can be used to estimate the futility of continuing to look for a physiological difference that likely does not exist.

The sample size for the cohort was selected for convenience; no statistical method was used to predetermine sample size. With the sample size of this cohort, we anticipated being only able to detect large effects. Post-hoc calculations of the effect size for a phenotyping sample of 21 and 17 participants to achieve a power of 80% is 0.94. Similar calculations for the exercise sample of nine and six participants is 1.27.

Inherent in the data are numerous estimates of effect size and correlation, even for variables that do not reach statistical significance. While the precision of these effect sizes may be poor, they are reported

to provide a sense of the strength of relationship and would be useful for determining statistical power for future research.

Statistically, each measure in the protocol was analyzed independent from all others. The statistical testing approach for each measure is listed in Supplementary Data S1. Where appropriate, statistical correction for multiple comparisons was performed within the measurement analyzed. Given the exploratory nature of the study, no statistical correction for multiple comparisons was performed across the different measures or for correlational analyses.

For analysis of biological samples, two or more technical replicates were used. Clinical evaluations were performed without replication. There was no randomization in this study. There were no interventions in this study. All researchers performing the experimental analysis of biological samples were blinded to diagnostic group. Evaluating clinicians were not blinded to diagnostic group. Details about data exclusions for each analysis performed can be reviewed in Supplementary Data S23.

### Statistical analysis of heart rate variability
A text file indicating timing in milliseconds between normal beats was exported. Rstudio 1.1.463(19) was used to remove non-normal beats and re-order data start-time to 8am. The subsequent text file was imported into Kubios HRV Version 1.0 with an artifact correction threshold of 0.3 s. HRV analysis was performed as recommended by the 1996 ESC and NASPE HRV task force and European Heart Rhythm Association. No de-trending was performed. Time-domain metrics included NN interval, pNN50, RMSSD, SDNN, SDNNi and SDANN measured over day(12-h), night(12-h), and 24-h periods. Frequency-domain metrics included VLF($0$–$0.04$ Hz), LF($0.04$–$0.15$ Hz), and HF($0.15$–$0.4$ Hz) measured over day (12-h), night (12-h), 24-h, and 5-min periods. Frequency analysis was conducted using a Lomb-Scargle periodogram with a smoothing window width of 0.02 Hz. Non-linear metrics included SD1, SD2, SD1/SD2 and were measured in one hour segments. Representative traces (HR, LF, HF) were plotted using data sampled at one minute intervals.

Subsequent analyses between study participants and controls were performed using GraphPad Prism version 9.0.0 for Mac, (GraphPad Software, San Diego, California USA) and SAS 9.4 (SAS Institute, Cary, NC). Mann–Whitney tests and Chi-square tests were used to evaluate the difference between HV and PI-ME/CFS participants. Scatterplot graphs display a bar signifying the median of the distribution. Non-linear measures (SD1, SD2 and SD1/SD2) and heart rate were examined using a mixed-effects model to account for 24-h repeated measurements with adjustments for hour of day; results of these latter measures were reported and displayed as least-square mean (lsmean) +/− standard error (stand err.) for PI-ME/CFS and HVs, respectively. A $p$-value $< 0.05$ is considered statistically significant.

### Statistical analysis of effort expenditure for rewards task
Following the analytic strategy described by Treadway[15], generalized estimating equations (GEE) were used to model the effects of trial-by-trial and participant variables on hard task choice. A binary distribution and logit link function were used to model the probability of choosing the hard task versus the easy task. All models included reward probability, reward magnitude, expected value (the product of reward probability and reward magnitude), and trial number, in addition to binary categorical variables indicating participant group and sex. Emulating Treadway et al., the two-way interactions between PI-ME/CFS diagnosis and reward probability, PI-ME/CFS diagnosis and reward magnitude, and PI-ME/CFS diagnosis and expected value were also tested, as was the three-way interaction among PI-ME/CFS diagnosis, reward magnitude, and reward probability. One new two-way interaction, the interaction of PI-ME/CFS diagnosis and trial number, was tested as well in order to determine whether rate of fatigue differed by diagnostic group.

Departing from the procedures described by Treadway[15], GEE was also used to model the effects of trial-by-trial and participant variables on task completion. A binary distribution and logit link function were again used given the binary nature of the task completion variable (i.e., success or failure). The model included reward probability, reward magnitude, expected value, trial number, participant diagnosis, and participant sex, as well as a new term indexing the difficulty of the task chosen (easy or hard). The three-way interaction of participant diagnosis, trial number, and task difficulty was evaluated in order to determine whether participants' abilities to complete the easy and hard tasks differed between diagnostic group, and in turn whether fatigue demonstrated differential effects on probability of completion based on diagnosis and task difficulty. Additionally, GEE was used to model the effects of these independent variables and interactions on button press rate, to provide an alternative quantification of task performance. This time, the default distribution and link function were used. The model's independent variables and interaction terms were the same as in the above task completion model.

All three sets of GEE models were performed using an exchangeable working correlation structure. Unstructured models were tested as well, but failed to converge. All GEE models were implemented in SAS 9.4.

### Statistical analysis of repetitive grip testing
Grip force data was filtered with a lowpass 8 Hz butterworth of order 2 and normalized to the maximum voluntary contraction. To represent the evolution of fatigue, data collected during the 1st block was compared to the last successful block and the three following failed blocks.

### Statistical analysis of functional MRI repetitive grip testing
AFNI[124,125] was used to process anatomical and EPI timeseries with the afni_proc.py tool that included removing the first two volumes, despiking the timeseries, registering the EPI data to the anatomical scan, adjusting for slice timing offsets, motion correcting these timeseries referring to least outlier volume with rigid body transformations using cubic polynomial interpolation, and spatially blurring the timeseries with a 6 mm FWHM Gaussian kernel. @ANATICOR was used to remove white matter signal from the timeseries to reduce scanner-related artifacts[126] and also to remove CSF signal. The motion limit was set to 3 mm and removed volumes with more than 10% of outliers as defined with 3dToutcount tool in AFNI. The demeaned and derivatives of head motion parameters were regressed out. The anatomical and the EPI timeseries were transformed to the MNI template with non-linear transformations.

Regression analysis with AFNI's 3dDeconvolve tool was used, with a box car model for each 30 s block of grip force. The first 16 blocks of the task were divided into quartiles of four blocks each. Blocks were pooled together in this fashion to better estimate fatigue-related brain activation. We used a different approach than in the TMS session to represent the evolution of fatigue because we needed to pool blocks together to better estimate fatigue-related brain activation. For group analysis, linear mixed-effects (3dLME tool in AFNI) were used with two groups and four blocks and participants as a random factor. A voxel threshold of $p \leq 0.01$ and a cluster threshold of $p \leq 0.05$, $k \geq 65$ (multiple comparisons correction) was used. We also used a t-test with the 3dMEMA tool in AFNI to assess commonly activated areas.

### Statistical analysis of RNA sequencing data
RNA sequence data was obtained from the libraries using the bcl2fastq v.2.17; Illumina software. RNA sequences were subjected to quality control (FastQC, a quality control tool for high throughput sequence data and available online at: http://www.bioinformatics.babraham.ac.uk/projects/fastqc), and trimmomatic (https://github.com/timflutre/trimmomatic) to remove adapters, followed by alignment to the human genome (GRCh38) using STAR[127]. Gene expression levels were

quantified using featuresCounts[128]. DE analysis was performed on PBMC and muscle RNAseq data using limma[129], and genes with nominal *p*-value ≤ 0.05 were considered DE. BMI was used as a covariate in sex separated and combined cohorts. Pathway enrichment analysis was performed using the R package clusterProfiler[130], which uses the fisher test to determine statistical significance. Additionally, prior known protein-protein interactions for the DE genes were extracted from the STRING resource (https://string-db.org/)[131]. Protein-protein interactions with a confidence score of >0.7 were reported. The fold change information of the genes node in the PPI network are highlighted as red (for upregulated genes) and blue (for downregulated gene) color nodes in Cytoscape[132].

### Statistical analysis of aptamer-based proteomics, metabolomics, and lipidomics data

For these datasets, the data analysis is described below. Partial least squares discrimination analysis (PLSDA) (mixOmics; https://bioconductor.org/packages/mixOmics/) was used to analyze the data and calculate the variable importance in prediction (VIP) scores for all the metabolites measured. Briefly, variables measurements with missing values in >50% of the samples were removed. For the remaining variables, the missing values were imputed with half of the minimum measurements for the respective metabolite. All the variables with VIP score >1 were subset for either pathway level inference or were assessed individually for known functions, and their expression level between groups are shown in the selected heatmaps. The features with VIP > 1 were considered to be important because the squared sum of all VIP values is equal to the number feature and thus, the average VIP would be equal to 1[133].

### Statistical analysis of transposable element expression data

RNA sequencing data from the PBMCs was aligned to the human genome (GRCh38) using STAR (PMID: 23104886) to allow for multi-aligned sequences using the following criteria: winAnchorMultimapNmax 100 and outFilterMultimapNmax 100. The transposable elements gene transfer file (GTF) was generated from the UCSC genome database. Gene expression quantification was performed with the UCSC RepeatMasker GTF file using the TEtranscripts tool (PMID: 26206304). The gene expression count matrix was used and DEseq2 R package was used to perform DE analysis. The summary DE table is reported. In parallel, the RNAsequencing data from the PBMCs were aligned to the human genome (GRCh38) using bowtie2, and HERV elements were quantified with the author-provided GTF file using Telescope tool (PMID: 31568525). No transposable elements were quantified from the dataset using this tool.

### Statistical analysis of microbiome shotgun metagenomics data

Comparisons between samples were interrogated from SummarizedExperiment objects[134] constructed using the JAMSbeta pipeline of the JAMS_BW package. Ordination plots were made with the t-distributed stochastic neighbor embedding (t-SNE) algorithm using the uwot package in R (https://github.com/jlmelville/uwot) and the ggplot2 library. Permanova values were obtained using the adonis function of the vegan package, with 10,000 permutations and pairwise distances calculated using Bray−Curtis distance. Heatmaps were drawn using the ComplexHeatmap package[135]. For each feature, *p*-values were calculated using the Mann−Whitney−Wilcoxon U-test on PPM relative abundances for that feature in samples within each group.

### Reproducibility of data analyses

In an effort to promote open science and reproducibility, the final source data files were reanalyzed where appropriate and individual source data files and source code used to analyze and visualize those data have been provided where possible and are available at https://github.com/docwalitt/National-Institutes-of-Health-Myalgic-Encephalomyelitis-Chronic-Fatigue-Syndrome-Code-Repository.

### Reporting summary

Further information on research design is available in the Nature Portfolio Reporting Summary linked to this article.

## Data availability

The Map ME/CFS databank (accession code: https://www.mapmecfs.org/group/post-infectious-mecfs-at-the-nih) contains demographics, performance validity testing, patient reported outcomes, small nerve fiber measures, neuronal injury markers, clinical lab data, heart rate variability measures, orthostatic challenge data, psychological scales, body composition measures, extracellular flux measures of PBMCs, muscle fiber measures, actigraphy and strength measures, cardio-pulmonary exercise test data, whole room calorimetry, neuropsychological testing, biological measures of blood and cerebrospinal fluid, [51] Chromium release assay, metabolomics, proteomics, lipidomics, mitochondrial sequencing, and stool nuclear magnetic resonance spectroscopy data. Accessing Map ME/CFS data requires signing up for an account but otherwise access to the data is unrestricted (Creative Commons BY 4.0). PBMC gene expression data (GEO Accession viewer (nih.gov): Accession Code GSE251872), muscle gene expression data (GEO Accession viewer (nih.gov): Accession Code GSE245661), and proteomics data (GSE251790 - GEO DataSets - NCBI (nih.gov): Accession Code: GSE251790; (GSE254030 - GEO DataSets - NCBI (nih.gov): Accession Code GSE254030) is available at Gene Expression Omnibus (GEO) and stool shotgun metagenomic data (SRA Links for BioProject (Select 954397) - SRA - NCBI (nih.gov), Accession Code SRP467038) are available at Sequence Read Archive, which are all available at BioProject (Homo sapiens (ID 954397) - BioProject - NCBI (nih.gov): Accession Code PRJNA954397). All neurophysiology data from transcranial magnetic stimulation and functional magnetic resonance imaging experiments are available at Pennsieve (Deep phenotyping of Post-infectious Myalgic Encephalomyelitis-Chronic Fatigue Syndrome - Blackfynn Discover (pennsieve.io); DOI: 10.26275/ile7-wrsk). External datasets analyzed include GEO GSE13033 (GEO Accession viewer (nih.gov)) and GEO GSE156792 (GEO Accession viewer (nih.gov)). Source data are provided with this paper.

## Code availability

Codes for the bioinformatic analysis of shotgun metagenomics are available at: https://github.com/johnmcculloch/JAMS_BW. The JAMS compatible Kraken2 taxonomic classification database used for shotgun metagenomics is available at: https://hpc.nih.gov/~mccullochja/JAMSdb202201.tar.gz. RNA sequence quality control was performed with FastQC: http://www.bioinformatics.babraham.ac.uk/projects/fastqc, trimmomatic: https://github.com/timflutre/trimmomatic, and multiqc: https://multiqc.info/. Codes for differential gene expression analysis are available at: COVID-19_Transcriptomics/Differential_gene_expression.R at master·NHLBI-BCB/COVID-19_Transcriptomics·GitHub. Codes for Pathway enrichment analysis are available at: COVID-19_Transcriptomics/PathwayEnrichment_clusterProfiler.R at master·NHLBI-BCB/COVID-19_Transcriptomics·GitHub. Transposable Seqeunce analysis performed with STAR: https://github.com/alexdobin/STAR, samtools: https://github.com/samtools/samtools, and TEcount tool: (https://github.com/mhammell-laboratory/TEtranscripts. Additionally, all the above codes and additional R scripts used in this study are available at: https://github.com/docwalitt/National-Institutes-of-Health-Myalgic-Encephalomyelitis-Chronic-Fatigue-Syndrome-Code-Repository.

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

## Acknowledgements

This research was supported (in part) by the Intramural Research Program of the NIH, including National Institute of Neurological Diseases and Stroke (NINDS), National Heart, Lung and Blood Institute (NHLBI), National Institute of Mental Health (NIMH), National Institute of Allergy and Infectious Disease (NIAID), National Institute of Diabetes, Digestion, and Kidney Disease (NIDDK), National Cancer Institute (NCI), National Institute of Aging (NIA), National Institute of Arthritis and Musculoskeletal and Skin Diseases (NIAMS), National Institute on Drug Abuse (NIDA), National Institute of Dental and Craniofacial Research (NIDCR), National Institute of Environmental Health Sciences (NIEHS), National Institute of Nursing Research (NINR), National Center for Complementary and Integrative Health (NCCIH), and NIH Clinical Center (CC). Grant support for this project included: ZIA NS003157 (A.N.), ZIA MH002922-14 (J.S.), ZIA HL 006210 (M.L.), ZIA HL 006212 (M.L.), ZIA DK071014 (K.C.), ZIA DK071014 (K.C.), ZIA HL005199 (M.S.), ZIA ES103362 (G.A.M.), and the NIH Common Fund (A.N.). We want to thank Francis Collins and Walter Koroshetz for their support of this project. Additionally, we would like to acknowledge the following people for their advice and support for this project: Anthony Komaroff, Lucinda Bateman, Benjamin Natelson, Andy Kogolnik, Daniel Peterson, Michael Tierney, Camilo Toro, Jeffery Lewis, Ana Acevedo, Jeffery Cohen, Nicolas Grayson, Fred Gill, Wendy Henderson, Nicolaas Fourie, Rosario Jaime-Iara, Paule Joseph, Eugene Major, Adriana Marques, Bonnie Hodsdon, Susan Robertson, Leora Comis, Dardo Tomasi, Neil Young, John Tsang, Rose Hayden, Olga Carlson, John Butman, Dima Hammoud, Govind Bhagvatheeshwaran, Eleanor Goulden, Renkui Bai, Michael Polydefkis, Jessica Gill, Chen Lai, Tracey Rouault, Manik Ghosh, and Angela Walitt.

## Author contributions

B.W., M.H., S.J., K.C., Y.E.A, R.A., R.J.B., A.W.B., P.D.B., L.M.K.C., A.B.C., B.D., L.F., D.S.G., S.G.H., M.L., J.J.L., A.L.M., J.A.M., J.D.M., L.B.R., M.N.S., L.N.S., B.S, J.S., B.J.S., G.T, S.A.T., S.B.Y., CHI Consortium, and A.N. conceived and designed the study. B.W., S.R.L., P.B., R.J.B., B.Cal., S.C., J.C., L.M.K.C., B.W.C., A.B.C., M.S.D., B.D., A.G., D.S.G., S.C.H., S.G.H., A.J.G., K.M.K., J.D.K., N.Mad., P.M.M., A.M., T.P., L.B.R., B.S., J.S., S.Si., S.So., B.J.S., S.A.T., C.S.V., F.V., C.V., A.W., S.B.Y, and A.N. recruited participants and collected clinical data, research data, and samples. B.W., S.R.L, Y.E.A., P.B., R.J.B., A.W.B., P.D.B., B.Cal., B.Cat., L.C., S.C., L.M.K.C., B.W.C., A.B.C., B.D., L.R.F., S.A.G., A.G., D.S.G., S.H., S.C.H., S.G.H., K.M.K., M.L., N.Mad., N.Mal., P.M.M., R.M., S.M.B., G.N., K.P., I.P-F., T.P., B.A.S., S.Si., J.S., B.J.S., S.A.T., C.S.V., F.V., C.V., A.W., S.B.Y, and CHI Consortium arranged and prepared samples and/or data for analysis. B.W., K.S., S.R.L., M.H., S.J., K.C., Y.E.A., J.J.B., P.B., R.J.B., A.W.B., P.D.B. B.Cal., B.Cat., L.C., F.C., L.M.K.C., A.B.C., B.D., L.R.F., D.S.G., S.C.H., S.G.H., K.R.J., K.M.K., M.L., N.Mad., N.Mal., A.L.M., J.A.M., P.M.M., R.M., G.A.M., S.M.B., G.N., I.P-F., M.N.S., F.S., S.Si., J.S., B.J.S., G.T., S.A.T., C.S.V., C.V, S.B.Y, CHI Consortium, and A.N. performed statistical analyses. B.W., K.S., S.R.L., M.H., S.J., K.C., R.A., P.B., R.J.B., F.C., L.M.K.C., B.D., L.F., D.S.G., S.G.H., M.L., N.Mad., A.L.M., J.A.M., G.A.M., M.N.S., J.S., B.J.S., C.V., CHI Consortium, and A.N. drafted the manuscript. All authors contributed to the revision and editing of the manuscript.

## Competing interests

The authors declare no competing interests.

## Additional information

Brian Walitt[1], Komudi Singh [2], Samuel R. LaMunion [3], Mark Hallett [1], Steve Jacobson[1], Kong Chen [3], Yoshimi Enose-Akahata [1], Richard Apps[4], Jennifer J. Barb [5], Patrick Bedard [1], Robert J. Brychta [3], Ashura Williams Buckley[6], Peter D. Burbelo [7], Brice Calco[1], Brianna Cathay[8], Li Chen[9], Snigdha Chigurupati[10], Jinguo Chen[4], Foo Cheung[4], Lisa M. K. Chin[5], Benjamin W. Coleman[11], Amber B. Courville [3], Madeleine S. Deming[5], Bart Drinkard[5], Li Rebekah Feng [12], Luigi Ferrucci [13], Scott A. Gabel[14], Angelique Gavin[1], David S. Goldstein [1], Shahin Hassanzadeh[2], Sean C. Horan [15], Silvina G. Horovitz[1], Kory R. Johnson [1], Anita Jones Govan[1], Kristine M. Knutson [1], Joy D. Kreskow[16], Mark Levin [2], Jonathan J. Lyons [17], Nicholas Madian [18], Nasir Malik[1], Andrew L. Mammen [19], John A. McCulloch [20], Patrick M. McGurrin [1], Joshua D. Milner [21], Ruin Moaddel[13], Geoffrey A. Mueller[14], Amrita Mukherjee[4], Sandra Muñoz-Braceras[19], Gina Norato[1], Katherine Pak[19], Iago Pinal-Fernandez [19], Traian Popa [1], Lauren B. Reoma[1], Michael N. Sack[2], Farinaz Safavi[1,17], Leorey N. Saligan [16], Brian A. Sellers[4], Stephen Sinclair[6], Bryan Smith[1], Joseph Snow[6], Stacey Solin[5], Barbara J. Stussman[1,18], Giorgio Trinchieri[20], Sara A. Turner[5], C. Stephenie Vetter [22], Felipe Vial[23], Carlotta Vizioli [1], Ashley Williams[24], Shanna B. Yang[5], Center for Human Immunology, Autoimmunity, and Inflammation (CHI) Consortium*, Avindra Nath [1] ✉

[1]National Institute of Neurological Diseases and Stroke (NINDS), Bethesda, MD, USA. [2]National Heart, Lung and Blood Institute (NHLBI), Bethesda, MD, USA. [3]National Institute of Diabetes, Digestion, and Kidney Disease (NIDDK), Bethesda, MD, USA. [4]NIH Center for Human Immunology, Autoimmunity, and Inflammation (CHI), Bethesda, MD, USA. [5]NIH Clinical Center (CC), Bethesda, MD, USA. [6]National Institute of Mental Health (NIMH), Bethesda, MD, USA. [7]National Institute of Dental and Craniofacial Research (NIDCR), Bethesda, MD, USA. [8]Texas A&M School of Engineering Medicine, College Station, TX, USA. [9]Affiliated Hospital of North Sichuan Medical College, Sichuan, China. [10]George Washington University Hospital, District of Columbia, Washington, DC, USA. [11]Allegheny General Hospital, Pittsburgh, PA, USA. [12]National Institute on Drug Abuse (NIDA), Bethesda, MD, USA. [13]National Institute of Aging (NIA), Baltimore, MD, USA. [14]National Institute of Environmental Health Sciences (NIEHS), Chapel Hill, NC, USA. [15]Sidney Kimmel Medical College, Philadelphia, PA, USA. [16]National Institute of Nursing Research (NINR), Bethesda, MD, USA. [17]National Institute of Allergy and Infectious Disease (NIAID), Bethesda, MD, USA. [18]National Center for Complementary and Integrative Health (NCCIH), Bethesda, MD, USA. [19]National Institute of Arthritis and Musculoskeletal and Skin Diseases (NIAMS), Bethesda, MD, USA. [20]National Cancer Institute (NCI), Bethesda, MD, USA. [21]Columbia University Medical Center, New York, NY, USA. [22]University of Colorado School of Medicine, Boulder, CO, USA. [23]Clínica Alemana Universidad del Desarrollo, Santiago, Chile. [24]Oakland University William Beaumont School of Medicine, Rochester, NY, USA. *A list of authors and their affiliations appears at the end of the paper. ✉e-mail: Avindra.nath@nih.gov

## Center for Human Immunology, Autoimmunity, and Inflammation (CHI) Consortium

Richard Apps[4], Jinguo Chen[4], Foo Cheung[4], Amrita Mukherjee[4] & Brian A. Sellers[4]

