## [Peer Review File · Nature Communications]

Deep phenotyping of Post-infectious Myalgic
Encephalomyelitis/Chronic Fatigue SyndromeEditorial Note: This manuscript has been previously reviewed at another journal that is not operating a transparent peer review scheme. This document only contains reviewer comments and rebuttal letters for versions considered at *Nature Communications*. Mentions of prior referee reports and the other journal have been redacted.

REVIEWER COMMENTS

Reviewer #3 (Remarks to the Author):

[REDACTED].

This study represents one of the most important and detailed investigations into a cohort of subjects with ME/CFS.

[REDACTED], the key results of this deep phenotyping study are the identification in ME/CFS cases of chronic antigenic stimulation with an increase in naïve and decrease in switched memory B-cells; the identification of brain dysfunction involving catechol pathways that potentially lead to the perception of fatigue and exercise intolerance; significant reproducible differences between male and female cases across multiple parameters; differences in immune cells, metabolites, and neurotransmitters in the cerebrospinal fluid; and significant differences between cases and controls in heart rate parameters. The reviewed manuscript further supports these key findings.

Although the sample size is unfortunately small because of restrictions imposed by the COVID pandemic, the cohort was meticulously screened to be as homogeneous as possible giving added credibility to the results. Importantly, these data may help guide other investigators as they further resolve the psychophysiology of ME/CFS.

[REDACTED].

I have carefully reviewed the revised manuscript in the context of my own suggestions as well as the suggestions of the other reviewers, and it is my opinion that the authors have reasonably and sufficiently addressed any issues put forth by the reviewers. I have no further comments and support the manuscript being published as soon as possible.

Reviewer #4 (Remarks to the Author):

[REDACTED].

I have a few remaining suggestions.

This small study did not find cognitive abnormalities (in attention, psychomotor speed/reaction time) or in NK cell function, yet these have been repeatedly confirmed by multiple laboratories in studies involving exponentially more subjects than participated in this study. The authors now acknowledge this, citing a few of the NK cell papers (but none of the cognitive studies). They propose that the difference between their findings and those of others are likely explained by the fact that prior studies did not

select cases quite as carefully as they did, and that this biased prior studies to find biologic abnormalities where none existed. Personally, I don't find this argument persuasive, and I think it disparages many excellent investigators. However, the authors obviously have the right to make this argument.

The figures display a number of comparisons between the cases and the healthy controls in which there are substantial quantitative differences that apparently do not achieve statistical significance. For example, the higher level of GFAP, a neuronal injury marker (Fig. S1G). If the authors do not want to report power calculations for each of these differences, the text could at least acknowledge that the study may have been underpowered to recognize some of the quantitatively great but statistically nonsignificant differences as "real".

I think the expanded language in the Discussion regarding altered effort preference might be unclear for many readers of a general scientific journal, like Nature Communications. I think the report would have greater impact if the authors eliminated jargon and explained some concepts that may be familiar to neurophysiologists but foreign to some readers. For example, the following two sentences (p. 10): "This difference in performance correlated with decreased activity of the right temporal-parietal junction, a part of the brain that is focused on determining "mismatch"³¹. In respect to movement, this would relate to the degree of agency³²." Mismatch between what, effort and reward? And what does "agency" refer to in this context?

[REDACTED]

On page 3 (Cohort Characteristics) I suggest that the words "laboratory tests" be replaced by "standard clinical laboratory tests".

Anthony L. Komaroff, MD

Reviewer #5 (Remarks to the Author):

[REDACTED]. The authors have attended to the concerns I raised in my previous review, either by adequately addressing them or providing appropriate justifications for their decisions.

Reviewer #6 (Remarks to the Author):

Summary: The study utilized a multi-disciplinary approach to investigate the underlying mechanisms and identify group differences in Post-infectious Myalgic Encephalomyelitis/Chronic Fatigue Syndrome (PI-ME/CFS). The study used a relatively homogeneous PI-ME/CFS population with post-infection symptom

onset. Volunteers underwent diverse physiological, cognitive, biochemical, microbiological, and immunological testing of blood, cerebrospinal fluid, muscle, and stool. Novel techniques measured physical capacity, effort preference, and deconditioning. Multi-omic analysis of gene expression, proteins, metabolites, and lipids was performed. The statistical approach used a broad, deep phenotyping with an exploratory design to generate new hypotheses. Strict case criteria and adjudication minimized misattribution. The analysis used a modified consilience concept, selecting measures to probe immunologic, bioenergetic, and homeostatic physiology facets. Significant differences were identified between PI-ME/CFS and healthy groups in immune function, metabolism, and autonomic function. Potential biomarkers were identified. Results are reported as HV mean \pm SD versus PI-ME/CFS mean \pm SD, p-value. The odds and relative odds ratios are reported as HV: PI-ME/CFS ratios [95% CI]. The study highlighted the need for further research to understand PI-ME/CFS pathogenesis.

Overall Evaluation: The article is ambitious and extensive analysis were conducted in a relatively small sample of subjects. The article is dense and difficult to read and crammed with details. It is difficult to elicit key take-home points that can be applied to clinical settings. The impact of the paper is likely to be limited to specialized settings such as research entities.

[REDACTED]

**REVIEWER COMMENTS**

We wish to thank the reviewers for their careful review of our manuscript and for their
insightful and helpful comments. We have addressed each of the comments in a pointwise
manner and made changes to the manuscript accordingly. Changes to the manuscript are listed
below in italic font.

**Reviewer #3 (Remarks to the Author):**

[REDACTED].

This study represents one of the most important and detailed investigations into a cohort of
subjects with ME/CFS.

[REDACTED], the key results of this deep phenotyping study are the identification in ME/CFS
cases of chronic antigenic stimulation with an increase in naïve and decrease in switched
memory B-cells; the identification of brain dysfunction involving catechol pathways that
potentially lead to the perception of fatigue and exercise intolerance; significant reproducible
differences between male and female cases across multiple parameters; differences in immune
cells, metabolites, and neurotransmitters in the cerebrospinal fluid; and significant differences
between cases and controls in heart rate parameters. The reviewed manuscript further
supports these key findings.

Although the sample size is unfortunately small because of restrictions imposed by the COVID
pandemic, the cohort was meticulously screened to be as homogeneous as possible giving
added credibility to the results. Importantly, these data may help guide other investigators as
they further resolve the psychophysiology of ME/CFS.

[REDACTED].

I have carefully reviewed the revised manuscript in the context of my own suggestions as well
as the suggestions of the other reviewers, and it is my opinion that the authors have reasonably
and sufficiently addressed any issues put forth by the reviewers. I have no further comments
and support the manuscript being published as soon as possible.

**Response** Thank you

**Reviewer #4 (Remarks to the Author):**

[REDACTED].

I have a few remaining suggestions.

This small study did not find cognitive abnormalities (in attention, psychomotor speed/reaction

time) or in NK cell function, yet these have been repeatedly confirmed by multiple laboratories
in studies involving exponentially more subjects than participated in this study. The authors
now acknowledge this, citing a few of the NK cell papers (but none of the cognitive studies).
They propose that the difference between their findings and those of others are likely
explained by the fact that prior studies did not select cases quite as carefully as they did, and
that this biased prior studies to find biologic abnormalities where none existed. Personally, I
don't find this argument persuasive, and I think it disparages many excellent investigators.
However, the authors obviously have the right to make this argument.

**Response:** Thanks for drawing our attention to the issue of cognitive dysfunction and NK cell
function. We have added some primary reference citations and modified the language:

*Neurocognitive, page 11:*

*This diverges from published data that suggests that small, heterogenous deficits in*
*performance can be demonstrated³⁹ which may not be evident in our study due to the small*
*sample size.*

*NK cell function, page 11:*

*This diverges from published data that suggests that NK cell function is decreased in ME/CF⁴⁸⁻⁵²,*
*which may not be evident in our study due to the small sample size.*

The figures display a number of comparisons between the cases and the healthy controls in
which there are substantial quantitative differences that apparently do not achieve statistical
significance. For example, the higher level of GFAP, a neuronal injury marker (Fig. S1G). If the
authors do not want to report power calculations for each of these differences, the text could
at least acknowledge that the study may have been underpowered to recognize some of the
quantitatively great but statistically nonsignificant differences as "real".

**Response:** We agree that the issue of effect size is an important one. We had addressed this in
the Supplement but have now added a sentence to the Limitations section in the main text to
emphasize this point on page 13:

*Post-hoc calculations of the effect size, for a phenotyping sample of 21 and 17 volunteers, to*
*achieve a power of 80% is 0.94, suggesting only large effects will be noted to be statistically*
*significant.*

I think the expanded language in the Discussion regarding altered effort preference might be
unclear for many readers of a general scientific journal, like Nature Communications. I think the
report would have greater impact if the authors eliminated jargon and explained some
concepts that may be familiar to neurophysiologists but foreign to some readers. For example,
the following two sentences (p. 10): "This difference in performance correlated with decreased
activity of the right temporal-parietal junction, a part of the brain that is focused on
determining "mismatch"³¹. In respect to movement, this would relate to the degree of

agency³².” Mismatch between what, effort and reward? And what does “agency” refer to in
this context?

**Response:** Thank you for pointing out where we could make the explanation more accessible.
These sentences, page 10, now read:

*This difference in performance correlated with decreased activity of the right temporal-parietal*
*junction, a part of the brain that is focused on determining “mismatch” between willed action*
*and resultant movement³¹. Mismatch relates to the degree of agency, the sense of control of the*
*movement³².*

[REDACTED]

**Response:** It may not be apparent that this noted neutrophil activation was not found in blood
or cerebrospinal fluid. Interestingly, however, a neutrophil pathway was found activated only in
muscle of men. Further, we did not find neutrophils in the muscle tissue on histological
evaluation. Hence, this is likely due to overlap with other genes in other pathways.

On page 3 (Cohort Characteristics) I suggest that the words “laboratory tests” be replaced by
“standard clinical laboratory tests”.

**Response:** Change made on page 3

*****

Anthony L. Komaroff, MD

**Reviewer #5 (Remarks to the Author):**

[REDACTED]. The authors have attended to the concerns I raised in my previous review, either
by adequately addressing them or providing appropriate justifications for their decisions.

**Response:** Thank you.

**Reviewer #6 (Remarks to the Author):**

Summary: The study utilized a multi-disciplinary approach to investigate the underlying
mechanisms and identify group differences in Post-infectious Myalgic
Encephalomyelitis/Chronic Fatigue Syndrome (PI-ME/CFS). The study used a relatively
homogeneous PI-ME/CFS population with post-infection symptom onset. Volunteers
underwent diverse physiological, cognitive, biochemical, microbiological, and immunological
testing of blood, cerebrospinal fluid, muscle, and stool. Novel techniques measured physical

capacity, effort preference, and deconditioning. Multi-omic analysis of gene expression,
proteins, metabolites, and lipids was performed. The statistical approach used a broad, deep
phenotyping with an exploratory design to generate new hypotheses. Strict case criteria and
adjudication minimized misattribution. The analysis used a modified consilience concept,
selecting measures to probe immunologic, bioenergetic, and homeostatic physiology facets.
Significant differences were identified between PI-ME/CFS and healthy groups in immune
function, metabolism, and autonomic function. Potential biomarkers were identified. Results
are reported as HV mean \pm SD versus PI-ME/CFS mean \pm SD, p-value. The odds and relative
odds ratios are reported as HV: PI-ME/CFS ratios [95% CI]. The study highlighted the need for
further research to understand PI-ME/CFS pathogenesis.

Overall Evaluation: The article is ambitious and extensive analysis were conducted in a
relatively small sample of subjects. The article is dense and difficult to read and crammed with
details. It is difficult to elicit key take-home points that can be applied to clinical settings. The
impact of the paper is likely to be limited to specialized settings such as research entities.

**Response:** We realize that the manuscript is dense. This is the most extensive study done to
date on ME/CFS. It was a massive undertaking involving nearly all Institutes in the NIH
intramural program. We created several teams of investigators each headed by a well-
recognized expert in the field. The manuscript represents the breadth and depth of the study.
We have now modified the discussion and conclusion sections to simplify the language, clearly
state the take home points for the clinicians taking care of these patients, listed the potential
therapeutic targets and we have a summary figure that shows the proposed pathways involved
in the pathogenesis of the syndrome and how each of the components studied are
interconnected. We have edited the final two paragraphs on pages 13-14 to emphasize the key
take home points.

*Clinically, this model suggests places for potential therapeutic intervention and why other*
*therapies have failed. The finding of possible immune exhaustion suggests that immune*
*checkpoint inhibitors may be therapeutic by promoting clearance of foreign antigen. Immune*
*dysfunction leads to neurochemical alterations that impact neuronal circuits, which may be*
*another point of intervention. Therapeutically targeting downstream mechanisms, with*
*exercise, cognitive behavioral therapy, or autonomic directed therapies may have limited impact*
*on symptom burden as it would not address the root cause of PI-ME/CFS. However, combination*
*therapy affecting multiple pathways could be considered. The finding of substantial*
*physiological differences related to sex suggest that there may not be a single unified*
*mechanism that leads to PI-ME/CFS and that successful therapy may ultimately require a*
*personalized medicine approach.*

*In conclusion, PI-ME/CFS is a distinct entity characterized by somatic and cognitive complaints*
*that are centrally mediated. Fatigue is defined by effort preferences and central autonomic*
*dysfunction. There are distinct sex signatures of immune and metabolic dysregulation which*
*suggest persistent antigenic stimulation. Physical deconditioning over time is an important*
*consequence. These findings identify novel therapeutic targets for PI-ME/CFS.*

[REDACTED]

Response: This paper was designed to compare the well-described cohorts to each other to determine if there were substantial group differences. In the subsequent study, several unbiased approaches will be used, with an emphasis on linkages between the many parameters identified in this manuscript to be scientifically relevant. With the breadth and depth of the data collected, the analysis required would be exhaustive and best presented in a separate manuscript.

[REDACTED]

Response: The finding of M1, cerebellum, and putamen activation was done via a conjunction analysis in which we took each group t-test across all blocks, thresholded voxels at $p \leq 0.01$ with a multiple comparison correction at $p = 0.05$, $k > 65$ and kept voxels that were commonly activated in each group.

In Supplement Table 4, line 20, we have added: *t-test*.

In Supplementary Methods, page 40, line 1811, we have added:

We also used a t-test with the 3dMEMA tool in AFNI to assess commonly activated areas.

In the manuscript text, page 5, we have augmented the paragraph explaining the fMRI processing in more detail:

First, we assessed commonly activated brain areas by implementing a conjunction analysis in which we took each group t-test across all blocks, thresholded voxels at $p \leq 0.01$ with a multiple comparison correction at $p = 0.05$, $k > 65$ and kept voxels that were commonly activated in each group. HV and PI-ME/CFS volunteers showed force-related brain activation in the left M1, right cerebellum, and left putamen during the task. We next assessed group differences with t-test (at $p = 0.01$, $k > 65$), but there was no difference between the groups. We also assessed changes across blocks with a two-way ANOVA (2 groups x 4 blocks) which showed that blood oxygen level dependent (BOLD) signal of PI-ME/CFS volunteers decreased across blocks bilaterally in temporo-parietal junction (TPJ) and superior parietal lobule, and right temporal gyrus in contradistinction to the increase observed in HVs ($F(3,45) = 5.4$, voxel threshold $p \leq 0.01$, corrected for multiple comparisons $p \leq 0.05$, $k > 65$; Figures 3J and 3K).

[REDACTED].

Response: The results presented come from an analysis at the whole brain level which is a common method in fMRI. There was no *a-priori* area selected, as this analysis was performed to explore what happened in the brain that led to the failure in performance. The TPJ is the result

of the analysis. There is extensive literature on the role of the TPJ in volition and perception of
the self and about the relationship between a willed action and the perception of a produced
movement, so the findings are not necessarily surprising. The authors could not know *a-priori*
whether the PI-ME/CSF group would fail to perform or not and whether the failure to perform
was due to central or peripheral causes. The experiment and analysis were designed to find a
region that would show differences wherever they might be.

We have revised the sentence with the reverse inference on page 5 of the manuscript: It now
reads:

*TPJ activity is inversely correlated with the match between willed action and the produced*
*movement.*¹

We have also added a sentence to the Functional MRI Repetitive Grip Testing section of
Supplementary Methods (page 31):

*This task was designed to identify, at the whole brain level, brain areas involved in “fatigue”.*

[REDACTED].

**Response:** The approach selected with GEE was necessary to determine the primary objective
of our study, the existence of EffRT performance difference between the PI-ME/CFS and HV
groups. The Cooper 2019 approach is not designed to determine group differences in
performance. Rather, it is designed to dissect out how participants are making their decisions
(i.e. which aspects of the task are being weighed in making decisions about hard/easy task
selection). Use of the Cooper 2019 approach would help determine the contribution of
individual aspects of the task to the performance outcome, such as how subjects integrate
reward, effort, and probability to guide decision-making. As our data did not show differences
in reward sensitivity and probability sensitivity by group, this approach seems unlikely to
provide information regarding the primary outcome. We have added this sentence to the
Supplement, page 10, lines 419-420:

*As Models 2 and 3 did not show differences in reward sensitivity and probability sensitivity by*
*group, further analysis was not performed*⁶.

We have also justified the reporting of trial timeouts with the following sentence in the
Supplement, page 10, lines 421-422:

*No difference in decision timeliness was observed as measured by task decision timeouts (0.3%*
*versus 0.6%, $p = 0.19$).*

1. Nahab FB, Kundu P, Gallea C, et al. The neural processes underlying self-agency. *Cereb Cortex* 2011;21(1):48-55. DOI: 10.1093/cercor/bhq059.

REVIEWERS' COMMENTS

Reviewer #4 (Remarks to the Author):

The authors have responded satisfactorily to my suggestions on the prior version. This new version will be a valuable contribution to the literature.

Reviewer #6 (Remarks to the Author):

The authors have responded to my concerns. I have no further comments.